# Increasing numerical stability of mountain valley glacier simulations: implementation and testing of free-surface stabilization in Elmer/Ice

André Löfgren[1], Thomas Zwinger[2], Peter Råback[2], Christian Helanow[1,3], and Josefin Ahlkrona[1,3]

[1]Department of Mathematics, Stockholm University, Sweden
[2]CSC — IT Center for Science Ltd, Espoo, Finland
[3]Swedish e-Science Research Centre, Sweden

**Correspondence:** André Löfgren (andre.lofgren@math.su.se)

**Abstract.** This paper concerns a numerical stabilization method for free-surface ice flow called the free-surface stabilization algorithm (FSSA). In the current study, the FSSA is implemented into the numerical ice-flow software Elmer/Ice and tested on synthetic two-dimensional (2D) glaciers, as well as on the real-world glacier of Midtre Lovénbreen, Svalbard. For the synthetic 2D cases it is found that the FSSA method increases the largest stable time-step size at least by a factor of five for the case of a gently sloping ice surface ( 3°), and by at least a factor of two for cases of moderately to steeply inclined surfaces ( 6° to 12°) on a fine mesh. Compared with other means of stabilization, the FSSA is the only one in this study that increases largest stable time-step sizes when used alone. Furthermore, the FSSA method increases the overall accuracy for all surface slopes. The largest stable time-step size is found to be smallest for the case of a low sloping surface, despite having overall smaller velocities. For an Arctic type glacier, Midtre Lovénbreen, the FSSA method doubles the largest stable time-step size, however, the accuracy is in this case slightly lowered in the deeper parts of the glacier, while it increases near edges. The implication is that the non-FSSA method might be more accurate at predicting glacier thinning, while the FSSA method is more suitable for predicting future glacier extent. A possible application of the larger time-step sizes allowed for by the FSSA is for spin-up simulations, where relatively fast changing climate data can be incorporated on short time scales, while the slowly changing velocity field is updated over larger time scales.

## 1 Introduction

Ice sheets and glaciers are important constituents of the global climate system and the mass loss from these is expected to be a main contributor to future sea-level rise (DeConto and Pollard, 2016; Hock et al., 2019; Meredith et al., 2019; Fox-Kemper et al., 2021). In order to reliably estimate future sea-level rise the accurate representation of ice-sheet and glacier dynamics is crucial, and higher-order physics models have proven to be instrumental in increasing the confidence in predictions (Hanna et al., 2013; Shepherd and Nowicki, 2017; Pattyn, 2018).

The most accurate description for the flow of ice, in the sense that all stress components are present in the Cauchy stress tensor, are the Stokes equations (Greve and Blatter, 2009). Approximations to the model are made by neglecting various

components, with some of the most notable examples being the shallow-ice approximation (SIA) (Hutter, 1983; Morland, 1984) and the first-order Stokes approximation (FOS) (Blatter, 1995; Pattyn, 2003). Owing to its simplicity and computational efficiency, the SIA method has a long history of use (see e.g., Blatter et al., 2010, for a historical overview); however, the SIA method has been found to be insufficient at reproducing the flow at regimes with a steep sloping bedrock (Meur et al., 2004; Dukowicz et al., 2011; Leng et al., 2012), as well as for smaller glaciers with a complex bedrock topography (Zwinger et al., 2007).

It has been demonstrated for lower-order physics models of shear dominated flow (such as SIA) that when coupled to the free-surface equation governing the evolution of ice sheets and glaciers they are subject to a parabolic type time-step size constraint that is highly dependent on the ice-domain thickness (Bueler et al., 2005; Gong et al., 2017; Bueler, 2022; Robinson et al., 2022). However, for the Stokes equations the same type of time-step size restriction does not necessarily hold true — even for setups where the SIA and the Stokes equations give qualitatively similar solutions (Löfgren et al., 2022). Still, for ice-sheet simulations using the Stokes equations, the restriction on the time-step size to have numerical stability, herein broadly defined as the unbounded growth of numerical errors, are typically found to be in the order of 0.1 to 10 yr (Gong et al., 2017; Löfgren et al., 2022). This is considerably smaller than typical time-scales at which ice-sheets evolve (Hindmarsh and Payne, 1996), which can be as large as 10000 yr (Greve and Blatter, 2009). Computation times can thus be cut if time-step sizes can be increased beyond the largest stable time-step size (LST) without compromising the desired accuracy of the solution.

One way of stabilizing the problem is to use a fully implicit time-stepping scheme, which has been demonstrated by Bueler (2016) for the SIA in a frozen-bed setting. However, the Stokes equations are considerably more expensive to solve than the simpler SIA equations and, since the nonlinear Stokes equations would have to be solved multiple times in each time step, makes such a scheme computationally infeasible for long-term simulations. Instead Kaus et al. (2010) propose the free-surface stabilization algorithm (FSSA) which modifies the weak formulation of the Stokes equations in order for the free-surface coupled system to mimic an implicit time-stepping scheme. The method was originally developed for mantle-convection simulations where a similar viscous-flow problem is solved, and multiple studies have indeed demonstrated that the method lengthens the LST substantially (Kaus et al., 2010; Duretz et al., 2011; Kramer et al., 2012; Andrés-Martínez et al., 2015; Rose et al., 2017).

From a glaciological perspective, a limitation of the original FSSA method is that linear rheologies are used on domains that are geometrically isotropic, meaning that they span equally in horizontal and vertical directions; i.e., domain aspect ratios are 1:1. A notable exception is Glerum et al. (2020), which considers both a similar shear-thinning nonlinear rheology and domains with aspect ratios in the order of 1:10. Still, the values of the physical parameters describing ice flow are different from that in mantle convection, and aspect ratios of ice sheets can be as small as 1:1000.

These issues were addressed by Löfgren et al. (2022), where the FSSA method was adapted to ice-flow modeling. It was concluded that the method works well in an ice-dynamical setting, and for the problems presented showed the potential to increase the LST by an order of magnitude. Nevertheless, one of the shortcomings in this case is that the method was only applied to simple ice-sheet benchmark problems, and more complex glacier simulations, e.g., using variable bedrock topography and sliding conditions, were only touched on briefly in the supplementary material and not studied thoroughly.

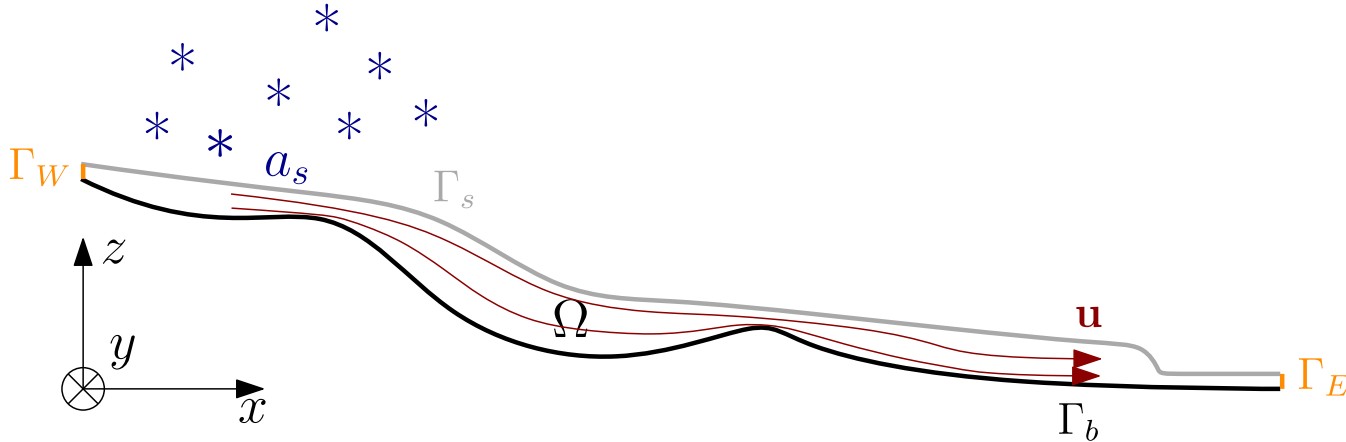

**Figure 1.** Cross section of a generic glacier domain $\Omega$ with its bedrock, $\Gamma_b$, marked in black and its surface, $\Gamma_s$, marked in gray. The west, $\Gamma_W$, and east, $\Gamma_E$, sides are marked in orange. The dark-red lines in the interior are the flow lines of the velocity field $\mathbf{u}$, and $a_s$ is the accumulation/ablation rate.

This work focuses on addressing these issues and applying the FSSA method to the regime of glacier modeling, considering both slip conditions and steep bedrock and surface inclinations. The method is assessed with regards to stability and accuracy for a synthetic case in two dimensions (2D) with a random-generated bedrock topography, using a novel method based on so-called Perlin noise (Perlin, 1985) and a real-world application to the glacier of Midtre Lovénbreen, Svalbard. The experiments are carried out using the ice-sheet solver Elmer/Ice (Gagliardini et al., 2013), where the FSSA method has been implemented.

The rest of the paper is structured as follows: in Sect. 2 the equations governing the flow of ice are presented; Sect. 3 introduces the numerical methods, including a presentation of the FSSA method for ice-sheet and glacier simulations; in Sect. 4, the experiments are presented along with their results; and finally the paper is concluded in Sect. 5 with a discussion of the results and the general outlook of the FSSA method from the perspective of glacier modeling.

## 2 Governing equations

### 2.1 The Stokes equations

The dynamics of ice flow can be described as a very slow-moving gravity-driven highly viscous fluid and is as such governed by the Stokes equations, see e.g., Greve and Blatter (2009)

$$\nabla \cdot (2\eta(\mathbf{u})\dot{\varepsilon}(\mathbf{u})) - \nabla p = \rho g \hat{\mathbf{z}}, \quad \mathbf{x} \in \Omega, \tag{1}$$

$$\nabla \cdot \mathbf{u} = 0, \quad \mathbf{x} \in \Omega, \tag{2}$$

where Eq. (1) follows from conservation of momentum and Eq. (2) from the conservation of mass. Furthermore, $\dot{\varepsilon} = \frac{1}{2}\left(\nabla \mathbf{u} + \nabla \mathbf{u}^T\right)$ is the strain-rate tensor, $\mathbf{u}$ and $p$ are the ice velocity and pressure, respectively, at spatial coordinate $\mathbf{x}$ in the domain $\Omega \subset \mathbb{R}^d$

(see Fig. 1), where $d \in \{2,3\}$ is the geometrical dimension. Furthermore, $\rho = 910$ kg m$^{-3}$ is the ice density and $g = 9.8$ m/s$^2$ is the acceleration due to gravity. Lastly, $\eta$ is the effective viscosity, which for ice depends on the velocity and temperature through Glen's flow law (Glen, 1955; Nye, 1957)

$$\eta(\mathbf{u}, T') = A(T')^{-\frac{1}{n}} \left( \frac{1}{2} tr(\dot{\varepsilon}^2) + \dot{\varepsilon}_0^2 \right)^{\frac{1-n}{2n}}. \tag{3}$$

Here $n = 3$ (Cuffey and Paterson, 2010) is the Glen or power-law exponent and $\dot{\varepsilon}_0^2 = 10^{-10}$ yr$^{-2}$ is a small regularization term
added in order to avoid an infinite viscosity at zero strain rates. The rate factor $A(T')$ depends on the ice temperature relative to the pressure melting point $T'$ through the Arrhenius equation (Glen, 1955)

$$A(T') = A_0 \exp \left( -\frac{Q}{RT'} \right), \tag{4}$$

where $A_0 = 2.89165 \times 10^{-13}$ s$^{-1}$ Pa$^{-3}$ is a pre-exponential factor, $Q = 60$ kJ mol$^{-1}$ is the activation energy and $R = 8.314462$ JK$^{-1}$mol$^{-1}$ is the ideal gas constant. The values stated here are the ones recommended by Paterson (1994) for ice at a temper-
ature $T' \geq -10°$ C when $n = 3$.

## 2.2   Boundary conditions

In order to specify appropriate boundary conditions (BC), the glacier boundary, $\partial \Omega$, is divided into non-overlapping boundary parts $\Gamma_s$, $\Gamma_W$, $\Gamma_b$ and $\Gamma_E$, see Fig. 1. The ice surface $\Gamma_s$ is the only non-stationary part of the domain, meaning the future of the glacier is determined purely by the evolution of the surface. For the different parts of the boundary, the following BCs are
considered

$$\sigma \hat{\mathbf{n}} = \mathbf{0}, \quad \mathbf{x} \in \Gamma_s, \tag{5}$$

$$\mathbf{u} \cdot \hat{\mathbf{n}} = 0, \quad \mathbf{x} \in \partial \Omega / \Gamma_s, \tag{6}$$

$$\hat{\mathbf{t}}_i \cdot \sigma \hat{\mathbf{n}} = -\beta |\mathbf{u}|^{m-1} \mathbf{u} \cdot \hat{\mathbf{t}}_i, \quad \mathbf{x} \in \Gamma_b^s, \tag{7}$$

$$\mathbf{u} = \mathbf{0}, \quad \mathbf{x} \in \Gamma_b^f, \tag{8}$$

where $\sigma = 2\eta\dot{\varepsilon} - pI$ ($I$ is the identity matrix) is the Cauchy stress tensor, $\hat{\mathbf{n}}$ is the unit normal outward pointing to the boundary, $\{\hat{\mathbf{t}}_i\}_{i=0}^{d-1}$ are tangent vectors spanning the plane defined by $\hat{\mathbf{n}}$, $\beta \geq 0$ is the drag coefficient, and $m \geq 1$ is an exponent.

  The explanation of each BC is as follows: Eq. (5), is a stress-free condition on the glacier surface, following from the assumption that the stresses asserted on the surface due to, for instance, wind or the atmospheric pressure are negligible compared to the internal stresses (Greve and Blatter, 2009). The second BC, Eq. (6), is an impenetrability condition under
which ice cannot flow into the bedrock, meaning its velocity in the direction normal to the bedrock must necessarily be zero. The third BC, Eq. (7), is a Weertman-type sliding law (Weertman, 1957), stating that the ice may slip along the bedrock, following a power law relation between the slip velocity and the shear stress. This study focuses only on the case for which $m = 1$, such that the relation is linear. Lastly, the fourth BC, Eq. (8), is a no-slip BC representing conditions where the ice is frozen to bedrock. The bedrock thus consists of parts, $\Gamma_b^s$, where slip is present and parts, $\Gamma_b^f$, where no slip occurs.

## 2.3 The free-surface equation

The time evolution of a glacier (or an ice sheet) is determined by its surface position $z_s = z_s(x,y,t)$ and is governed by a separate equation called the free-surface equation (Greve and Blatter, 2009)

$$\frac{\partial z_s}{\partial t} + u_x^s \frac{\partial z_s}{\partial x} + u_y^s \frac{\partial z_s}{\partial y} = u_z^s + a_s, \tag{9}$$

where $a_s$ is the vertical rate of mass accumulation (or ablation), $\mathbf{u}^s = (u_x^s, u_y^s, u_z^s)$ is the velocity field from the Stokes equations Eq. (1) – (2) evaluated on the surface boundary $\Gamma_s$, see Fig. 1. Furthermore, the bedrock $z_b(x,y)$ is assumed to be impenetrable and rigid, such that the following constraint is fulfilled at all times $t$,

$$z_s(x,y,t) \geq z_b(x,y) + H_{min}, \tag{10}$$

where $H_{min}$ is the minimum ice thickness. Equation (10) together with the weak formulation of Eq. (9) forms a variational inequality, which is solved using a method of imposed Dirichlet conditions, as described in Gagliardini et al. (2013).

## 3 Computational aspects

### 3.1 Solution procedure

A first order time-stepping approach for solving the Stokes equations coupled to the free-surface equation is shown in Fig. 2a and consists of first solving the Stokes equations, Eq. (1) – (2), for the velocity field evaluated on the surface, $\mathbf{u}^s$, which then enters as coefficients into the free-surface equation, Eq. (9). The free-surface equation is then solved for a new height function, $z_s(x,y,t+\Delta t)$, which in turn determines the new domain $\Omega(t+\Delta t)$. The mesh is updated based on an extruded-mesh principle, wherein nodes are vertically aligned in columns such that the mesh can be updated by simply displacing nodes vertically according to the new height function $z_s(x,y,t+\Delta t)$; see e.g., Löfgren et al. (2022) for implementation details. This process is repeated until the final simulation time is reached. This is the standard approach in ice-sheet modeling (used in e.g., Elmer/Ice Gagliardini et al., 2013), and is in this study referred to as an explicit time-stepping scheme in terms of velocity.

This explicit time-stepping scheme can be contrasted with a Picard linearized implicit time-stepping scheme, with respect to the coupled system, updating both velocities and geometry simultaneously (Bueler, 2016, 2022). An example of a first order implicit scheme, available in Elmer/Ice, is shown in Fig. 2b, where an extra loop is needed in order to the resolve the velocity field $\mathbf{u}(t+\Delta t)$ over the next time step. This has the disadvantage that the computationally expensive nonlinear Stokes equations need to be solved repeatedly in each time step, by iterating back- and forth between the domain at the old time step $\Omega(t)$ and the domain at the new time step $\Omega(t+\Delta t)$. The advantage is that it is numerically stable, allowing for large time-step sizes. The goal of this paper is to evaluate an approach, the FSSA, that finds a solution which is close to the solution yielded by the implicit time-stepping scheme without adding the extra computationally costly iteration. It is thus an approach that uses the explicit time-stepping scheme in a way that is stable and without substantial loss of accuracy.

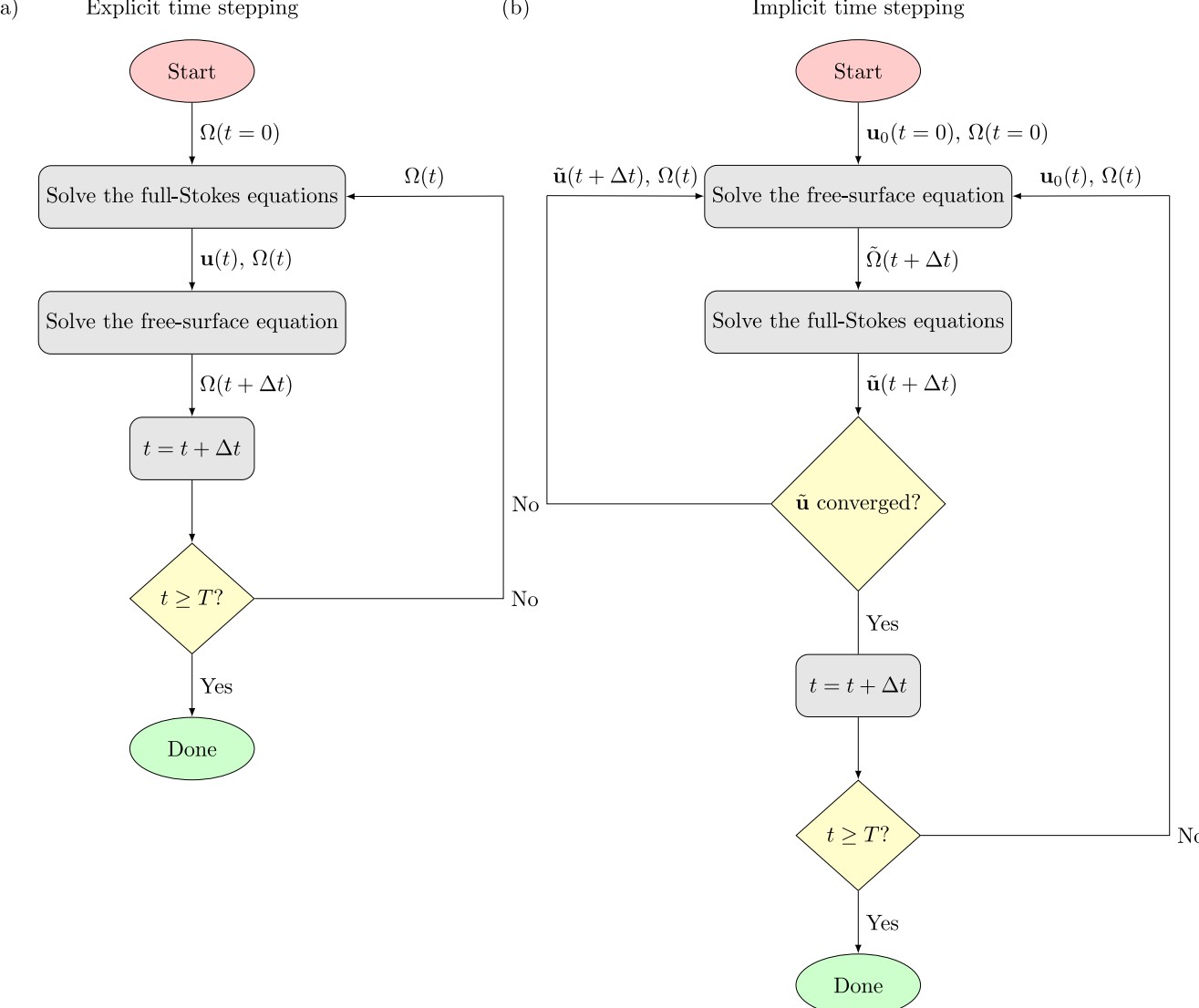

**Figure 2.** Examples of first order (a) explicit- (b) and implicit (Picard linearization) time-stepping schemes. For each time step the explicit scheme solves the Stokes equations, Eq. (1) – (2), for the velocity field $\mathbf{u}(t)$ on the domain of the current time step $\Omega(t)$, the solution $\mathbf{u}(t)$ then enters as a coefficient into the free-surface equation, Eq. (9), from which the domain at next time step, $\Omega(t + \Delta t)$, is obtained directly and the time is updated, i.e., $t = t + \Delta t$. The implicit scheme on the other hand solves for the velocity at the next time step $\mathbf{u}(t + \Delta t)$. The procedure for obtaining $\mathbf{u}(t + \Delta t)$ is as follows: in each time step an initial guess $\mathbf{u}_0(t)$ is inserted into the free-surface equation to obtain an estimate of the new domain $\tilde{\Omega}(t + \Delta t)$. The Stokes equations are then solved on $\tilde{\Omega}(t + \Delta t)$ to obtain an estimate $\tilde{\mathbf{u}}(t + \Delta t)$. This estimate is then checked if it is close to the initial guess $\mathbf{u}_0(t)$. If the estimate is not close, the process is repeated, using $\tilde{\mathbf{u}}(t + \Delta t)$ as the new initial guess, until convergence is obtained. Finally, after convergence, the time is updated. For both cases the algorithm terminates when the final simulation time, $T$, is reached.

## 3.2 The Stokes weak formulation — the basis for the stabilization method

The Stokes equations, Eq. (1) – (2), are discretized and solved numerically using the finite element method (FEM), which first requires recasting the problem in its weak form: Find $(\mathbf{u}, p) \in \mathcal{X} \times \mathcal{Q}$ such that

$$(\dot{\varepsilon}(\mathbf{v}) : 2\eta\dot{\varepsilon}(\mathbf{u}))_\Omega - ((\nabla \cdot \mathbf{v}), p)_\Omega - (q, \nabla \cdot \mathbf{u})_\Omega + (\mathbf{v}, \beta\mathbf{u})_{\Gamma_b} = -(\mathbf{v}, \rho g\hat{\mathbf{z}})_\Omega, \tag{11}$$

for all $(\mathbf{v}, q) \in \mathcal{X} \times \mathcal{Q}$. Here the colon operator $A : B$ between matrices $A$ and $B$ denotes their Frobenius inner product. Furthermore, $\mathcal{X}$ and $\mathcal{Q}$ are appropriate function spaces satisfying the so-called *inf-sup* stability condition (Ladyzhenskaya, 1969; Babuška, 1971; Brezzi, 1974). The fact that the forcing term is constant inside the integrals of the inner products opens up for the construction of the FSSA method, as will be described in the next section.

The nonlinear nature of Eq. (11) requires linearizing the viscous term with Picard or Newton iterations. Convergence issues sometimes prohibit using Newton solvers for glaciological problems. To overcome these issues relaxation methods can be employed.

## 3.3 Free surface time discretization and stabilization

The weak formulation of the Stokes-coupled free-surface equation, Eq. (9) – (11), is discretized in time by first evaluating all integrals in Eq. (11) at time $t = t^{k+\theta}$, so that the weak formulation reads: Find $(\mathbf{u}^{k+\theta}, p^{k+\theta}) \in \mathcal{X} \times \mathcal{Q}$ such that

$$(\dot{\varepsilon}(\mathbf{v}) : 2\eta\dot{\varepsilon}(\mathbf{u}^{k+\theta}))_{\Omega^{k+\theta}} - (\nabla \cdot \mathbf{v}, p^{k+\theta})_{\Omega^{k+\theta}} - (q, \nabla \cdot \mathbf{u}^{k+\theta})_{\Omega^{k+\theta}} + (\mathbf{v}, \beta\mathbf{u}^{k+\theta})_{\Gamma_b^{k+\theta}} = -(\mathbf{v}, \rho g\hat{\mathbf{z}})_{\Omega^{k+\theta}}, \tag{12}$$

for all $(\mathbf{v}, q) \in \mathcal{X} \times \mathcal{Q}$. Here $k$ denotes the time step and $\theta \in \mathbb{R}$ is an implicitness parameter for which $\theta = 0$ yields an explicit solver and $\theta = 1$ an implicit. Secondly, a semi-implicit Euler discretization is employed for the free-surface equation, Eq. (9), such that

$$z_s^{k+1} + \Delta t \left( (u_x^s)^{k+\theta} \frac{\partial z_s^{k+1}}{\partial x} + (u_y^s)^{k+\theta} \frac{\partial z_s^{k+1}}{\partial y} \right) = z_s^k + \Delta t \left( (u_z^s)^{k+\theta} + a_s^{k+1} \right) \tag{13}$$

where $\mathbf{u}_s^{k+\theta} = ((u_x^s)^{k+\theta}, (u_y^s)^{k+\theta}, (u_z^s)^{k+\theta})$ is the surface velocity obtained from Eq. (12), $\Delta t$ is the time-step size and $z_s^{k+1}$ is the unknown surface at time step $k+1$ to be solved for. The scheme is called semi-implicit since it is implicit in terms of the surface $z_s$ and explicit in terms of velocity $\mathbf{u}$ when $\theta = 0$.

The explicit nature, when $\theta = 0$, of the time-stepping scheme in Eq. (13) makes it prone to numerical instabilities. A fully implicit scheme ($\theta = 1$) would, on the other hand, involve a highly expensive computation in order to evaluate $\mathbf{u}^{k+1}$, see Fig. 2b. To circumvent this problem, the FSSA was introduced by Kaus et al. (2010) for mantle-convection problems and adapted to glaciological problems in Löfgren et al. (2022). The FSSA mimics a fully implicit scheme by approximating all integrals on the left-hand side in Eq. (12) by $\int_{\Omega^{k+\theta}} \cdot d\Omega \approx \int_{\Omega^k} \cdot d\Omega$, and estimating the force term over the new domain by applying Reynold's transport theorem (see e.g., Greve and Blatter, 2009)

$$\frac{d}{dt} \int_{\Omega(t)} f \, d\Omega = \int_{\Omega(t)} \frac{\partial f}{\partial t} \, d\Omega + \int_{\partial\Omega(t)} (\mathbf{u}_b \cdot \hat{\mathbf{n}}) \, f \, d\Gamma. \tag{14}$$

In ice-sheet modeling $\mathbf{u}_b = \mathbf{u} + a_s\hat{\mathbf{z}}$ is the velocity of the moving boundary and $f = -\rho g\hat{\mathbf{z}} \cdot \mathbf{v}$. Since $f$ is constant in time the first term on the right-hand side of Eq. (14) becomes zero. Taking this into account gives, using forward Euler, the estimate of the forcing on the time step $k + \theta$ as

$$(\mathbf{v}, \rho g\hat{\mathbf{z}})_{\Omega^{k+\theta}} \approx (\mathbf{v}, \rho g\hat{\mathbf{z}})_{\Omega^k} + (\theta\rho g\Delta t(\mathbf{v} \cdot \hat{\mathbf{z}}), (\mathbf{u} + a_s\hat{\mathbf{z}}) \cdot \hat{\mathbf{n}})_{\Gamma_s^k}. \tag{15}$$

Note how the integrals of the weak form and the fact that $\rho g$ is constant are features that opens up for using Reynolds transport theorem in this simple way.

Inserting Eq. (15) into the Stokes weak formulation Eq. (11), and approximating the integrals on the left-hand side by $\int_{\Omega^{k+\theta}} \cdot d\Omega \approx \int_{\Omega^k} \cdot d\Omega$, yields the FSSA stabilized weak formulation appropriate for glaciology: Find $(\tilde{\mathbf{u}}^{k+\theta}, \tilde{p}^{k+\theta}) \in \mathcal{X} \times \mathcal{Q}$ such that

$$(\dot{\varepsilon}(\mathbf{v}) : 2\eta\dot{\varepsilon}(\tilde{\mathbf{u}}^{k+\theta}))_{\Omega^k} - (\nabla \cdot \mathbf{v}, \tilde{p}^{k+\theta})_{\Omega^k} - (q, \nabla \cdot \tilde{\mathbf{u}}^{k+\theta})_{\Omega^k} + (\mathbf{v}, \beta\tilde{\mathbf{u}}^{k+\theta})_{\Gamma_b^k} + (\rho g\theta\Delta t\mathbf{v} \cdot \hat{\mathbf{z}}, \tilde{\mathbf{u}}^{k+\theta} \cdot \hat{\mathbf{n}})_{\Gamma_s^k}$$
$$= -(\mathbf{v}, \rho g\hat{\mathbf{z}})_{\Omega^k} - (\rho g\theta\Delta t a_s\mathbf{v} \cdot \hat{\mathbf{z}}, \hat{\mathbf{n}} \cdot \hat{\mathbf{z}})_{\Gamma_s^k}, \tag{16}$$

for all $(\mathbf{v}, q) \in \mathcal{X} \times \mathcal{Q}$. For the FSSA, letting $\theta = 1$, the velocity $\tilde{\mathbf{u}}^{k+1}$ is only an approximation of the solution $\mathbf{u}^{k+1}$ obtained from the fully implicit scheme. The validity of the FSSA follows from the fact that the gravitational force is driving the ice flow.

To better understand the effect of the stabilization term, insight can be gained by applying the FSSA to the SIA approximation of the Stokes equations, for which it can be shown that FSSA coincides approximately with evaluating the pressure at the end of the time integration (see the appendix in Löfgren et al. (2022)).

## 3.4 Weak formulation of the free-surface equation

The time-discretized free-surface equation, Eq. (13), is discretized spatially using FEM, which requires recasting it to its weak formulation: Find $z_s^{k+1} \in \mathcal{V}$ such that

$$(v, z_s^{k+1})_{\Gamma_s^\perp} + \Delta t\left(v, (\tilde{u}_x^s)^{k+\theta}\frac{\partial z_s^{k+1}}{\partial x}\right)_{\Gamma_s^\perp} + \Delta t\left(v, (\tilde{u}_y^s)^{k+\theta}\frac{\partial z_s^{k+1}}{\partial y}\right)_{\Gamma_s^\perp} = (v, z_s^k)_{\Gamma_s^\perp} + \Delta t\left(v, (\tilde{u}_z^s)^{k+\theta}\right)_{\Gamma_s^\perp} + \Delta t\left(v, a_s^{k+1}\right)_{\Gamma_s^\perp}, \quad \forall v \in \mathcal{V}, \tag{17}$$

where $\Gamma_s^\perp \subset \mathbb{R}^{d-1}$ is the projection of the free surface $\Gamma_s$ onto the underlying plane (or line in two dimensions) with $z = 0$. This advection-type equation is in Elmer/Ice stabilized by either the residual-free bubbles (RFB) method (Baiocchi et al., 1993) or streamline upwind Petrov-Galerkin (SUPG) stabilization (Franca and Frey, 1992). The stabilizing impact of the transport stabilization is investigated in this study.

## 4 Numerical experiments

### 4.1 Overview

In this section two experiments using varying bedrock slopes and sliding conditions are presented to demonstrate the applicability of the FSSA method to glacier modeling and to assess its stabilizing properties. In the first experiment, the method is applied to a 2D flow-line case, with an undulating bedrock generated using gradient noise (see Appendix A), superimposed on a sloping bedrock. The FSSA method is investigated with regards to accuracy and stability for different bedrock slopes, mesh resolution and upwinding. The second experiment applies the FSSA method to the real-world glacier of Midtre Lovénbreen, Svalbard, and is also evaluated based on stability and accuracy.

### 4.2 Experiment 1: 2D "Perlin" Glacier

#### 4.2.1 Setup: advancing glacier

This experiment consists of a 2D glacier geometry with a sloping, undulating bedrock, where accumulation and sliding conditions are present. The bedrock is generated by superimposing three gradient-noise octaves (see Appendix A) on a parabola, such that

$$z_b(x) = \frac{\alpha}{L_x}(x - L_x)^2 + C^1 \text{octave}_1(\Delta x^1, x) + C^2 \text{octave}_2(\Delta x^2, x) + C^3 \text{octave}_3(\Delta x^3, x) \tag{18}$$

where $\alpha$ is the average slope, $L_x = 8000$ m is the horizontal extent of the domain and the octaves represent noise of different frequencies. The coefficient of the parabola has been chosen such that $\frac{dz_b}{dx}(0) = 2\alpha$ and $\frac{dz_b}{dx}(L_x) = 0$. The noise amplitudes are set to $C^1 = 300$ m, $C^2 = 500$ m and $C^3 = 600$ m and the respective octave frequencies to $\Delta x^1 = 2000$ m, $\Delta x^2 = 1000$ m and $\Delta x^3 = 500$ m. The resulting bedrocks are visible in Fig 3.

The initial ice surface is a thin layer of ice

$$z_s(x, 0) = z_b(x) + 10 \text{ m}. \tag{19}$$

To build up a glacier on the bedrock, a non-negative accumulation function that is linearly decaying (with the horizontal coordinate) with a maximum at $x = 0$ is used:

$$a(x) = \max\left(1 - \frac{3x}{L_x}, 0\right). \tag{20}$$

The BCs imposed are impenetrability, Eq. (6), on the west- and east boundaries, $\Gamma_W$ and $\Gamma_E$ (see Fig. 1). On the surface $\Gamma_s$, the free-surface condition Eq. (5) is applied. Lastly, on the bedrock, $\Gamma_b$, the impenetrability BC, Eq. (6), is combined with the linear Weertman sliding law of Eq. (7), with a drag coefficient given by

$$\beta(x) = \beta_{min} + \frac{\beta_{max} - \beta_{min}}{1 + \exp\left(\frac{x - \mu}{\sigma}\right)}, \tag{21}$$

where $\beta_{max} = 1000$ MPa yr m$^{-1}$, $\beta_{min} = 0.01$ MPa yr m$^{-1}$, $\sigma = 200$ m and $\mu = 3000$ m. This drag coefficient should be viewed as a transition from a no-slip condition when $x \ll \mu$ to a free-slip condition when $x \gg \mu$, with the length of the transition zone controlled by $\sigma$.

Firstly, a study is performed to investigate how both stability and accuracy of the FSSA method are influenced by increasing bedrock slopes for an advancing glacier. Simulations are performed on three domains with different average bedrock slopes in Eq. (18): $\alpha = 0.05$ ($\approx 2.9°$) (gently sloping glacier), $\alpha = 0.1$ ($\approx 5.7°$) (moderately sloping glacier) and $\alpha = 0.2$ ($\approx 11.5°$) (steep sloping glacier).

To estimate the error, a reference solution is obtained for all three cases by performing simulations using a short time-step size $\Delta t = 0.05$ yr and a fine mesh resolution with $(N_x, N_z) = (1000, 10)$, where $N_x$ and $N_z$ are the number of layers in the horizontal- and vertical directions, respectively. The reference simulations are performed until final times $t = 900$ yr, $t = 700$ yr and $t = 500$ yr, for the respective cases. Subsequent simulations using coarser temporal resolutions are then started from an intermediate glacier surface obtained from the reference simulation, starting from times $t = 500$ yr, $t = 400$ yr, and $t = 300$ yr. The error is then estimated by comparing the ice thickness, $H(x) = z_s(x) - z_b(x)$, of the coarse solution to the thickness of the reference simulation, $H_{ref}$, at the final times for the respective slopes. The ice thickness error $\epsilon_H$ is computed as

$$\epsilon_H = \frac{||H_{ref} - H||_2}{||H_{ref}||_2}, \tag{22}$$

where $||\cdot||_2$ denotes the discrete L2-norm. The error is then measured for multiple coarse temporal resolution simulations, with and without FSSA, for various time-step sizes $\Delta t$.

A stability study is conducted estimating the LST for different mesh resolutions for the case $\alpha = 0.1$. Starting from an intermediate glacier surface obtained at time $t = 300$ yr, 15 time steps are performed using a fixed $\Delta t$. In each time step, stability is examined by calculating the infinity norm of the velocity field: the solver is said to be unstable if $||u_z||_\infty \geq 100$ m yr$^{-1}$. If the simulation is deemed stable, then $\Delta t$ is incremented by 1 yr and the simulation is restarted.

Following Gong et al. (2017) and Löfgren et al. (2022), all simulations use a constant rate factor $A = 100$ yr$^{-1}$MPa$^{-3}$ in Eq. (3) and a constant ice density $\rho = 910$ kg m$^{-3}$. The elements used to solve the Stokes equations are the *inf-sup* stable Taylor-Hood elements (Taylor and Hood, 1973), and RFB is used to stabilize the transport problem.

### 4.2.2 Setup: retreating glacier

A study is also conducted to compare the stabilizing impact of the FSSA method to the standard upwinding schemes RFB and SUPG, both of which are available in Elmer/Ice. In this case the methods are applied to a glacier subject to a negative net mass balance. Investigating such a case is of interest for two reasons: firstly, since the glacier approaches a steady state it allows for performing very long term simulations without the glacier reaching the end of the domain; secondly, a negative accumulation could potentially affect the stabilizing impact of the FSSA due to triggering the surface limiter imposing the minimum ice thickness, for which in this study the FSSA has not been adapted to take into consideration.

Stability is, as in the previous of the advancing glacier study, evaluated based on the relative increase of the LST as compared to not using FSSA. For this reason the same stability study as in previous experiment is performed for the three cases: no

upwinding, RFB, and SUPG. The starting glacier surface is the final surface obtained at time $t = 700$ yr of the reference simulation. All simulations use a mesh resolution of $(N_x, N_z) = (1000, 10)$.

To obtain a retreating glacier, melting is introduced into the accumulation function by simply letting

$$a(x) = 1 - \frac{3x}{L_x},\tag{23}$$

which is essentially the same accumulation function as Eq. (20) but allowing for negative values.

    In order for the glacier to experience sliding throughout the simulation, the drag coefficient is modified so that slip occurs predominantly in the interior of the domain

$$\beta(x) = \beta_{min} + \frac{\beta_{max} - \beta_{min}}{1 + \exp\left(\frac{x - \mu}{\sigma}\right)} + \frac{\beta_{max} - \beta_{min}}{1 + \exp\left(\frac{L_x - x - \mu}{\sigma}\right)},\tag{24}$$

where $\beta_{max} = 1000$ MPa yr m$^{-1}$, $\beta_{min} = 0.001$ MPa yr m$^{-1}$, $\sigma = 100$ m and $\mu = 1500$ m. Since this experiment is designed to study the effect of upwinding, the minimum drag coefficient is reduced by a factor of ten, compared to the previous advancing case, in order for the Stokes-coupled free-surface system to admit a more transport-like behavior, i.e., large horizontal velocities with a shear-to-slip ratio close to zero.

### 4.2.3   Results: advancing glacier

The reported estimated ice thickness error, $\epsilon_H$, as calculated by Eq. (22), are shown in Table 1 for different bedrock slopes $\alpha$ and FSSA stabilization parameters $\theta$. The FSSA method ($\theta = 1$) is seen to be stable for time-step sizes, $\Delta t$, up to 25 yr for $\alpha = 0.05$ and $\alpha = 0.1$, while for $\alpha = 0.2$ it was stable for all tested $\Delta t$, up to 50 yr. In the case of $\theta = 0$ the largest stable time-step size (LST) is between 5 to 10 yr for $\alpha = 0.05$ and $\alpha = 0.1$ and between 10 to 25 yr for $\alpha = 0.1$ and $\alpha = 0.2$. The FSSA

method thus increases the LST by at least a factor of two for all cases, and may even be as large as five times for $\alpha = 0.05$. Compared to ice-sheet simulations in e.g., Löfgren et al. (2022) the time-step sizes are large even without stabilization.

    In Table 2 the LST is reported for different mesh sizes for the intermediate sloping case $\alpha = 0.1$. It is seen that without the FSSA that the LST is by all practical means mesh independent, which is in agreement with the mesh studies carried out by Löfgren et al. (2022). On the other hand, for the FSSA a slight mesh dependence of $LST \sim \Delta x^{0.4}$ is seen. This means that the

stabilizing effect of FSSA decreases for higher mesh resolutions, where the relative increase in the LST reduces from 3 for the coarser mesh $(N_x, N_z) = (400, 10)$ to 2.1 for the finer mesh $(N_x, N_z) = (1000, 10)$.

    The LST is larger for the steep bedrock case, despite the velocity field also having a larger magnitude (see Fig. 3d–f). The reason for this might be related to the low and moderately inclined bedrock cases having a greater ice thickness, for which analytical expressions derived using zeroth-order approximation, e.g., SIA, have shown a strong inverse relation between

the LST and the ice thickness (Gong et al., 2017; Robinson et al., 2022). In Löfgren et al. (2022) it was shown that the characteristics of the instabilities for Stokes-coupled free-surface flow are related to the domain aspect ratio: thicker domains tend to give rise to long wavelength sloshing instabilities, while thin domains give rise to numerical oscillation of shorter wavelengths.

In the current experiment, thick domains are represented by $\alpha = 0.05$ and $\alpha = 0.1$ where velocities are low enough for

a thicker ice to develop, and the thin domain is represented by $\alpha = 0.2$. Indeed from Fig. 4, which shows a time series of vertical velocity profiles for unstable time-step sizes, it is seen that for low and moderate bedrock slopes, Fig. 4a–c and Fig. 4d–f, respectively, the instabilities behave differently than for the case of a steep bedrock, Fig 4g–i. In the former cases the instabilities emerge as result of the vertical velocity profiles shifting in sign and growing in magnitude between time steps, resulting in the glacier surface sloshing around the stable reference surface (black dashed lines in Fig 4). For the steep bedrock

case, while sloshing is possible to discern in the interior, the most prominent feature of the instabilities are the high frequency numerical oscillations occurring at the glacier front, seen in Fig. 4i. Despite the different characteristic of the instabilities arising for the different cases, it is clear from Table 1 and Fig. 3 that the FSSA method mitigates instabilities in both cases, as was also concluded by Löfgren et al. (2022).

Comparing the errors of the FSSA method ($\theta = 1$) to no stabilization ($\theta = 0$) in Table 1, it is seen that the FSSA method

generally yields a much more accurate solution, and as $\Delta t$ increases, the discrepancy grows in favor of the FSSA. For $\Delta t \geq 5$ yr the error for all slopes is almost twice as large when $\theta = 0$ compared to $\theta = 1$. For some cases, even using a twice as large time-step size with FSSA compared to no FSSA, the increase in the error is much less than the expected 100 %.

Generally, the error increases linearly with the time-step size $\Delta t$, i.e., $\epsilon_H = \mathcal{O}(\Delta t)$, as is expected of the semi-implicit Euler time-stepping scheme in Eq. (13), as long as stability restrictions are satisfied. This observation holds regardless of the slope $\alpha$

and stabilization parameter $\theta$.

Figure 3 shows the glacier surfaces at the final times for different $\Delta t$ and $\theta$. In all cases it is seen from the zoom-in plots that a too large time-step size gives a too thin glacier front, compared to the reference solution. Comparing Table 1 and Fig. 3, it is seen that a larger error corresponds to a thinner glacier front. Consequently, the FSSA method, which is generally more accurate in this experiment, yields a faster moving front for the same $\Delta t$. This is expected based on the fact that the FSSA

method is a quasi-implicit time-stepping scheme, meaning it uses an estimate of the velocity from the next time step to update the glacier surface in Eq. (9). Since the glacier is growing in size, the velocity field is expected to increase in magnitude over the duration of the simulation, such that $||\mathbf{u}^{k+1}|| \geq ||\mathbf{u}^k||$ ($k$ denotes the time step), meaning that the FSSA method gives a larger velocity coefficient in the free-surface equation, and thus yields a faster moving front.

The error also increases with the bedrock slopes $\alpha$, despite the simulation times being shorter, which is expected given that

larger slopes give a higher velocity coefficient in the free-surface equation, Eq. (9). From the large error seen for the steep bedrock case, $\alpha = 0.2$, it is not even clear that stability considerations are the limiting factor for the time-step size, it might as well in practical applications come down to accuracy — depending on whether Eq. (22) is deemed a satisfactory error metric. From a practical point of view, this has the implication that using a time-step size close to the LST may not be a wise strategy when employing the FSSA method as the error in the final ice thickness is quite large for, e.g., the stable case of $\Delta t = 50$

310 yr, especially for the long term simulations considered in this experiment. On the other hand, the smaller errors observed for $\alpha = 0.05$ and $\alpha = 0.1$, indicates that stability considerations are more important for the time-step size. Thus, for a given error tolerance, the FSSA method seems to offer the greatest potential for speed-up for simulating glaciers that are on top of bedrocks

with a topography that is gently to moderately inclined. Regardless of surface slope, it is obvious that the FSSA method gives not only a more stable solver, but also increases its accuracy.

In summary, the main finding is that the FSSA method allows for using larger time-step sizes by increasing both accuracy and stability, for all slopes angles investigated. However, the method seems to offer the greatest benefit for low to moderately sloping glaciers, where the time-step size seems to be mainly limited by stability considerations.

### 4.2.4 Results: retreating glacier

The LST for different numerical schemes and FSSA stabilization parameters $\theta$ is reported in Table 3. Comparing with the LST
in the advancing case (Table 2), which uses the RFB, it is seen that the LST in this case is about three times smaller. This may be explained by the larger velocities of 90 m yr$^{-1}$ (see Fig. 5a) compared to 20 m yr$^{-1}$ (see Fig. 3e). Despite this, it is interesting that the larger velocities do not seem to have an impact on the stability of the FSSA, and in fact the LST is larger in this case (39 yr compared to 27 yr). However, it should be noted that the FSSA method yields smaller velocities as $\Delta t$ increases, see Fig. 5, which for the case of $\Delta t = 39$ yr have decreased to 45 m yr$^{-1}$. The fact that the LST is larger in this case
demonstrates that the FSSA is also applicable to cases where the free surface is constrained by a minimum ice thickness.

  Furthermore, it is seen that upwinding alone appears to have negligible impact on the stability of the solver, despite surface velocities being dominated by sliding (see Fig. 5). However, combining the FSSA with upwinding increases the LST slightly for RFB ($\sim 12\%$) and substantially for SUPG ($\sim 60\%$), compared to FSSA alone. The best choice stability wise thus seems to be combining FSSA and SUPG.

It was also found for the large time-step sizes $\Delta t > 25$ yr allowed for by the FSSA, that non-zero surface velocities started arising in the deglaciated areas. However, these could effectively be mitigated by setting the accumulation part of the FSSA to zero in deglaciated areas. This did not compromise stability, nor accuracy as all simulations deemed stable were found to approach the same steady state.

**Table 1.** Relative error in the ice thickness, as defined by Eq. (22), for different time-step sizes $\Delta t$, FSSA parameter $\theta$, and bedrock slopes $\alpha$. The error is calculated after final times 900 yr, 700 yr and 500 yr for the respective slopes $\alpha = 0.05$, $\alpha = 0.1$ and $\alpha = 0.2$. Entries marked with an X are unstable cases.

| | Ice thickness error (%) | | | | | |
| --- | --- | --- | --- | --- | --- | --- |
| | $\alpha = 0.05$ | | $\alpha = 0.1$ | | $\alpha = 0.2$ | |
| $\Delta t$ (yr) | $\theta = 0$ | $\theta = 1$ | $\theta = 0$ | $\theta = 1$ | $\theta = 0$ | $\theta = 1$ |
| 0.5 | 0.15 | 0.14 | 0.25 | 0.22 | 0.61 | 0.48 |
| 1 | 0.27 | 0.22 | 0.49 | 0.38 | 1.24 | 0.72 |
| 2.5 | 1.04 | 0.80 | 2.27 | 1.58 | 7.07 | 3.60 |
| 5 | 2.79 | 1.80 | 6.89 | 3.69 | 18.5 | 8.43 |
| 10 | X | 3.81 | 15.3 | 8.03 | 34.4 | 15.7 |
| 25 | X | 9.15 | X | 15.2 | X | 24.9 |
| 50 | X | X | X | X | X | 34.0 |

**Table 2.** Largest stable time-step size (LST) with and without FSSA stabilization for different mesh resolutions $(N_x, N_z)$, where $N_x$ and $N_z$ denote the number of horizontal- and vertical layers respectively. The resolution for the reported LST is 1 yr, meaning that it is between 12 and 13 yr for e.g., the case $\theta = 0$ and $(N_x, N_z) = (400, 10)$. The last column shows the relative increase in the LST.

| | LST Perlin glacier (advancing) (yr) | | |
| --- | --- | --- | --- |
| $(N_x, N_z)$ | Without FSSA $\theta = 0$ | With FSSA $\theta = 1$ | $LST(\theta = 1)/LST(\theta = 0)$ |
| $(400, 10)$ | 13 | 39 | 3.0 |
| $(600, 10)$ | 12 | 33 | 2.8 |
| $(800, 10)$ | 13 | 30 | 2.3 |
| $(1000, 10)$ | 13 | 27 | 2.1 |

**Table 3.** Largest stable time-step size (LST) with and without FSSA stabilization for the numerical schemes: no upwinding, residual-free bubbles (RFB), and streamline upwind Petrov-Galerkin (SUPG). The resolution for the reported LST is 1 yr, meaning that it is between 3 and 4 yr for e.g., the cases $\theta = 0$. The last column shows the relative increase in the LST.

| | LST Perlin glacier (retreating) (yr) | | |
| --- | --- | --- | --- |
| Upwinding | Without FSSA $\theta = 0$ | With FSSA $\theta = 1$ | $LST(\theta = 1)/LST(\theta = 0)$ |
| None | 4 | 35 | 8.8 |
| RFB | 4 | 39 | 9.8 |
| SUPG | 4 | 55 | 13.8 |

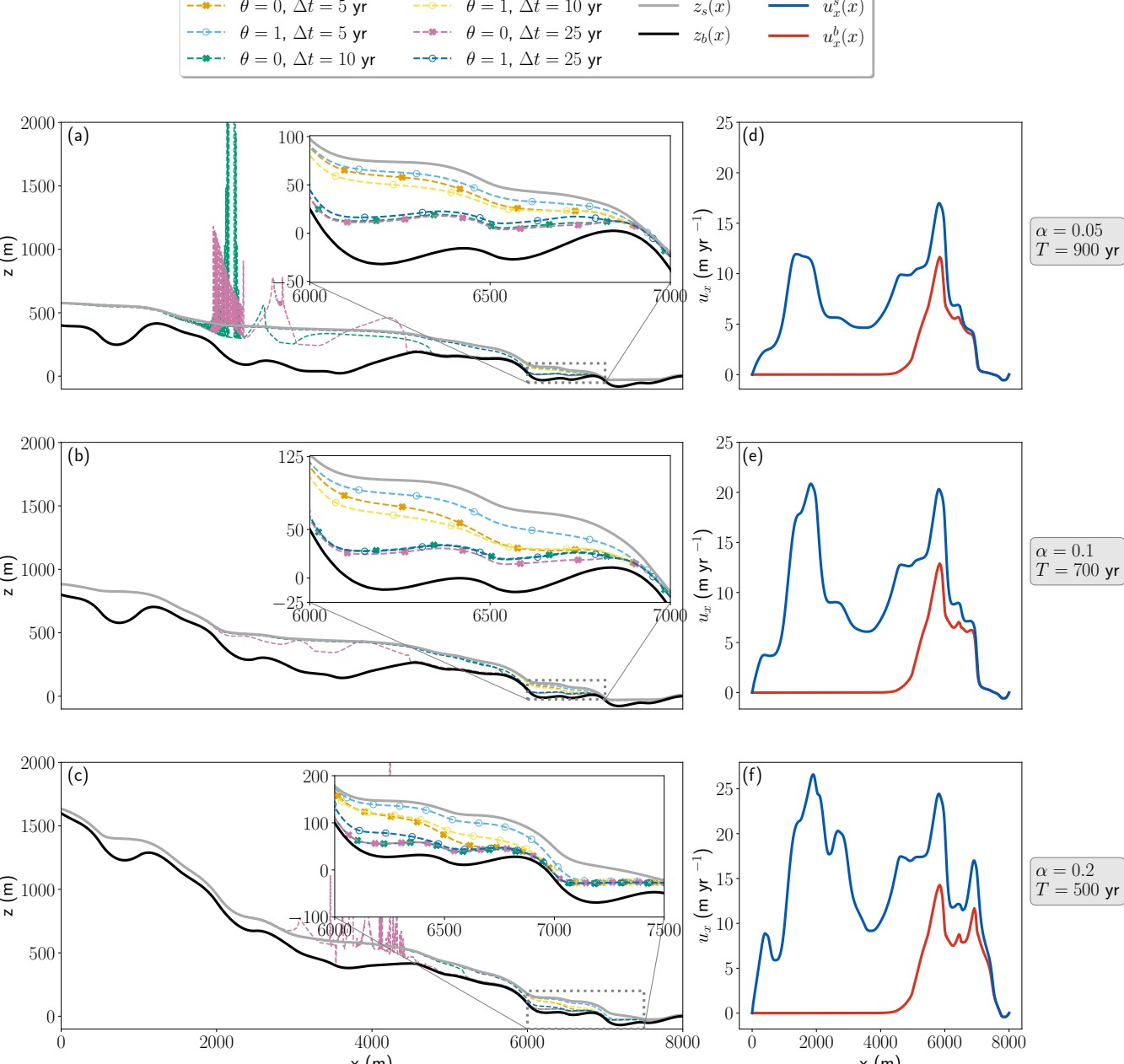

**Figure 3.** Glacier surfaces (a-c) and horizontal velocities of the reference simulations (d-f) at final times $T$ for different bedrock slopes $\alpha$, FSSA stabilization parameters $\theta$ and time-step sizes $\Delta t$. The gray solid lines in (a-c) are the reference surface, $z_s$, for the respective case, and the black solid lines the bedrock, $z_b$. The blue solid lines in (d-f) are the surface velocities, $u_x^s$, and the red solid lines the bedrock slip velocities, $u_x^b$. The reference solution was obtained using a short time-step size $\Delta t = 0.05$ yr.

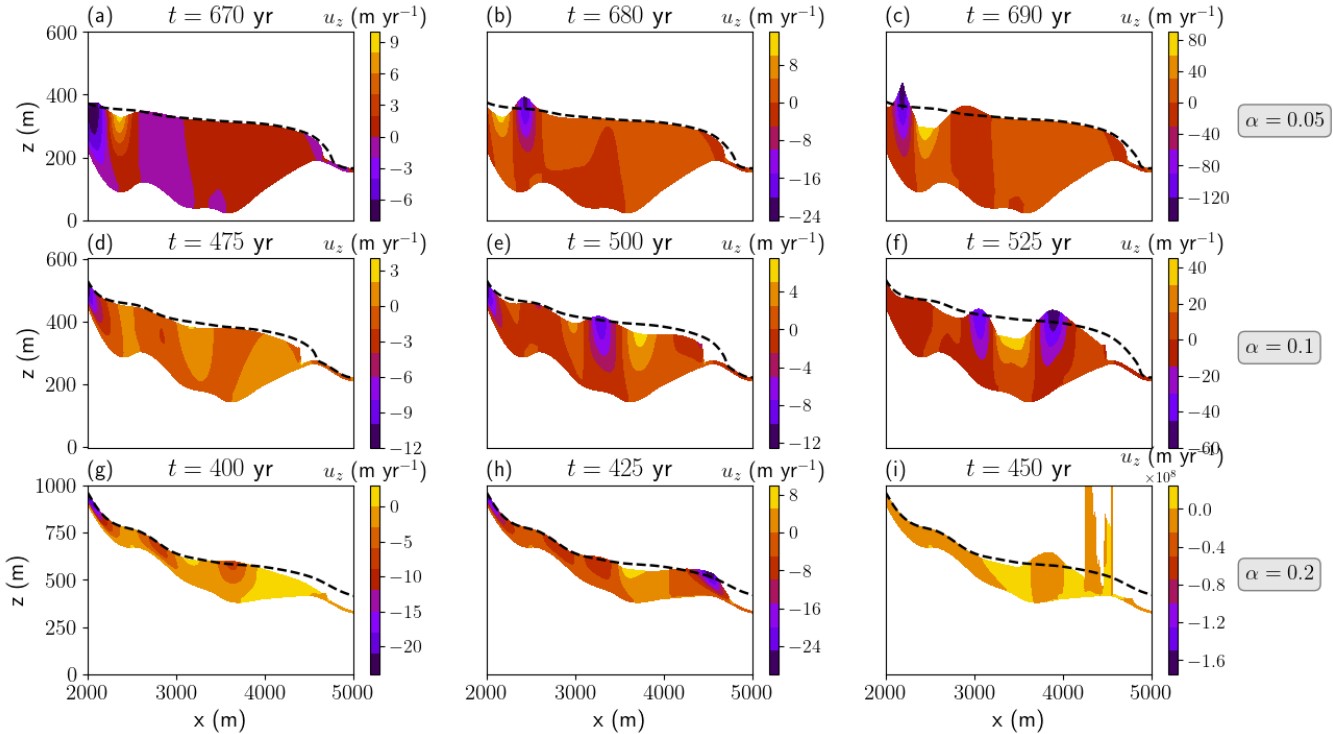

**Figure 4.** Vertical velocity profiles, $u_z$, from simulations using no FSSA ($\theta = 0$) and unstable time-step sizes $\Delta t = 10$ yr (a–c) and $\Delta t = 25$ yr (d–i) shown at the indicated times $t$. The profiles are shown for the different bedrock slopes $\alpha = 0.05$ yr (a–c), $\alpha = 0.1$ (d–f) and $\alpha = 0.2$ (g–i). The black dashed line in each figure is the glacier surface obtained from the stable reference simulation.

## 4.3 Experiment 2: Midtre Lovénbreen

### 4.3.1 Setup

This experiment aims to demonstrate how FSSA works for a real-world, three-dimensional glacier simulation. For this purpose, the valley glacier Midtre Lovénbreen, ($78.53°$N, $12.04°$ E) Svalbard, is chosen as it is a thoroughly studied glacier that has been modeled using Elmer/Ice previously (Zwinger and Moore, 2009; Välisuo et al., 2017). The glacier is in this study classified as an intermediate to steep sloping glacier with an average surface slope $\alpha = 0.14$ ($8°$). The initial geometry is shown in Fig. 6, where colors indicate ice thickness, Fig. 6a, surface mass balance (SMB), Fig. 6b, velocity magnitude, Fig. 6c, and gray contour lines represents surface heights in meters above sea level.

The basal BC is the linear sliding law of Eq. (7) with the same drag coefficient as in Välisuo et al. (2017)

$$\beta(x,y,t) = \begin{cases} 0.04 \text{ MPa yr m}^{-1} & \text{if } H(x,y,t) \geq 120 \text{ m,} \\ 10 \text{ MPa yr m}^{-1} & \text{if } H(x,y,t) < 120 \text{ m,} \end{cases} \tag{25}$$

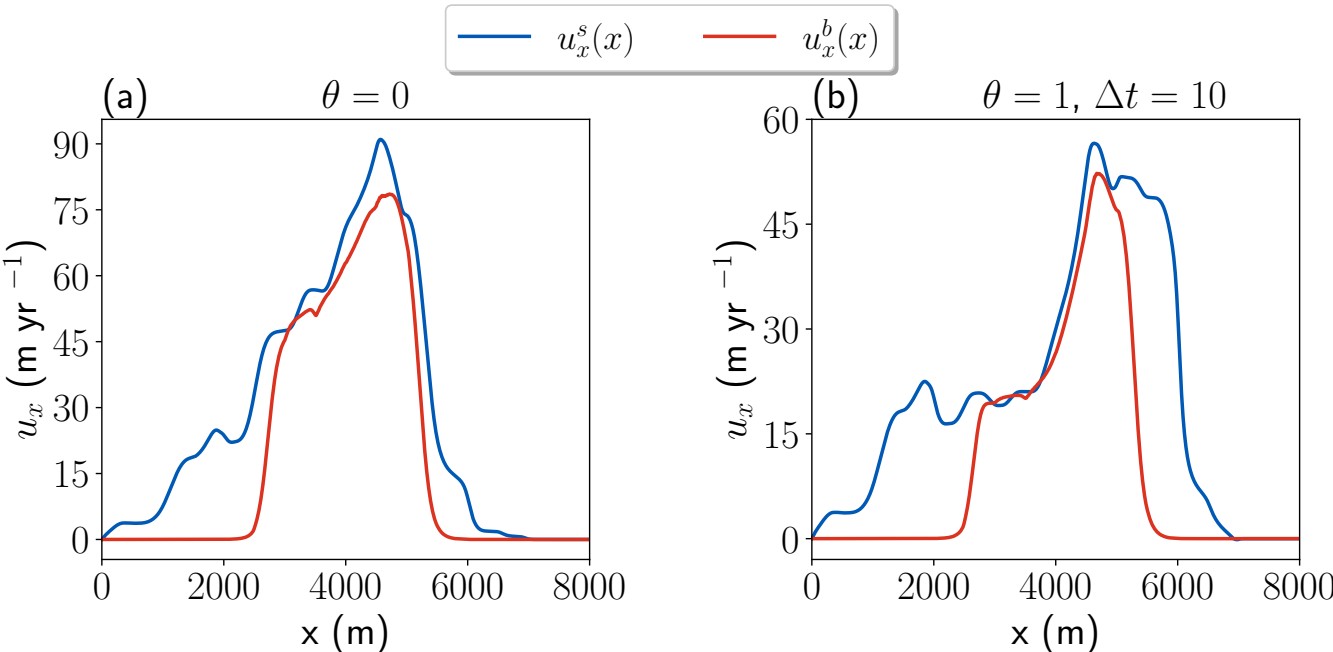

**Figure 5.** Horizontal velocity profiles (a) without FSSA and (b) with FSSA using stabilization parameter $\theta = 1$ and time-step size $\Delta t = 10$ yr. The blue solid line is the surface velocities, $u_x^s$, and the red solid line the bedrock slip velocities, $u_x^b$.

where $H$ is the ice thickness. This drag coefficient imposes high slip velocities at parts where the ice is thick ($\geq 120$ m)
and essentially imposes a no-slip condition at the shallower parts ($< 120$ m). Note that this also means that sliding velocities decrease as the glacier thins. The values for the drag coefficient were determined in Välisuo et al. (2017) by a manual inversion using observed surface velocities as input data. As in Välisuo et al. (2017), the bedrock elevation is given by a bedrock DEM created from ground-penetrating radar data (Rippin et al., 2003; Zwinger and Moore, 2009), the 1995 surface elevation DEM is based on digital photogrammetry from vertical aerial photographs, and the 2005 surface elevation DEM is a product derived
from airborne LIDAR (light detection and ranging) data (James et al., 2006).

Following Välisuo et al. (2017), the SMB used has been estimated from surface DEMs of the ice surface for the years 1995 and 2005. The SMB is obtained by solving the Stokes equations, Eq. (1) – (2), on the domain defined by the 1995 surface DEM, from the time-discretized free-surface equation, Eq. (13), the SMB is given by

$$a_s = \frac{z_s^{2005} - z_s^{1995}}{10 \text{ yr}} + u_x^{1995}\frac{\partial z_s^{2005}}{\partial x} + u_y^{1995}\frac{\partial z_s^{2005}}{\partial y} - u_z^{1995}. \tag{26}$$

The 3D mesh is generated by extruding a 2D footprint mesh into five layers ($N_z = 5$) with a horizontal resolution of about 25 m. The footprint is large enough to cover the glacier at the size it was in 1962, which implies that a large deglaciated area is included in the domain.

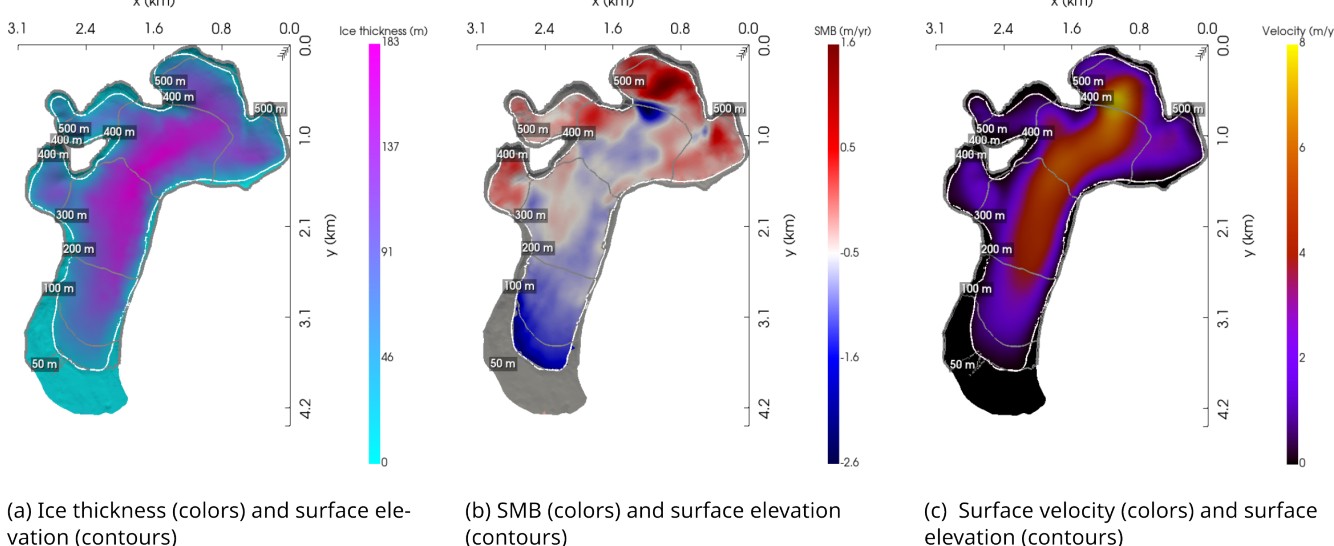

(a) Ice thickness (colors) and surface elevation (contours)

(b) SMB (colors) and surface elevation (contours)

(c) Surface velocity (colors) and surface elevation (contours)

**Figure 6.** Ice thickness, surface mass balance (SMB) excluding the deglaciated area (gray), modeled surface velocity and surface elevation of Midtre Lovénbreen at the start of the simulation, in year 1995. The ice flows from top to bottom. The computational domain is given by the 1962 extent of the glacier, the white line indicates the outline of the glacier at the start of the simulation. The figures were created using the open source visualization and analysis toolkit PyVista (Sullivan and Kaszynski, 2019).

For this experiment linear equal-order elements are used in conjunction with Galerkin least-squares stabilization (Hughes et al., 1986) to circumvent the *inf-sup* stability condition. Furthermore, the free surface is constrained by a minimum ice thickness of $H_{min} = 10$ m. The transport problem is stabilized using SUPG.

The glacier evolution is simulated from year 1995 to year 2195 with and without FSSA and compared and evaluated in regards to the LST and accuracy. To measure accuracy a reference solution is obtained by performing a simulation from year 1995 to 2195 using a small time-step size of $\Delta t = 1$ yr and no FSSA ($\theta = 0$). The error is estimated by comparing the ice thickness in the final solutions to the reference solution, for various time-step sizes. Stability is for this experiment evaluated qualitatively, based on the presence of spurious shifts in the sign of the vertical velocity, i.e. sloshing, since it was found that checking the norm of the velocity alone did not accurately predict the presence of instability in this case.

To remedy convergence issues of the Newton solver (which appeared both with and without FSSA) the derivative of the viscosity appearing in the Newton linearization is relaxed by a factor two-thirds.

### 4.3.2 Results

The final glacier surface after 200 years is shown in Fig. 7a, where the color denotes the ice thickness of the reference simulation. Comparing with the thickness of the initial glacier in Fig. 6a, it is seen that the glacier surface has retreated a distance of about 1.5 km uphill. In addition the glacier has also experienced thinning from initially having a maximum thickness of about 180 m down to a maximum thickness of 100 m. The retreat of the glacier is expected given the considerably negative SMB,

as can be seen from Fig. 6b. Note, however, that predicting the actual retreat of the glacier is not the objective of this study, but rather to demonstrate the stabilizing properties of the FSSA method for an SMB derived from experimental data. The true SMB is in reality likely to change substantially over the simulation period considered.

Figure 7a shows the glacier outlines for the stable reference simulation (white line), an FSSA stabilized simulation with $\Delta t = 40$ yr (dark gray line), and an unstabilized simulation with $\Delta t = 40$ yr (orange line). It is seen that the unstabilized case deviates substantially from the reference simulation, due to instability, while the glacier outline from FSSA simulation is to a large extent indistinguishable from the outline of the reference simulation. The largest time-step size tested for stability in each case is shown in Table 4 where it is seen that the FSSA simulation is stable for at least twice as large time-step sizes. From Fig. 7b it is seen, similarly to the 2D Perlin experiment, that instabilities arise in the deep parts of the domain, where the error is large — indicative of the sloshing-type instability encountered in the previous experiment. Indeed from Fig. 8d–f instability, in the sense of spurious shifts in the sign of the vertical velocity, for the case $\Delta t = 25$ yr and $\theta = 0$ is observed in the upper parts of the domain. These are effectively mitigated by reducing the time-step size (Fig. 8a–c) or by means of stabilization (Fig. 8g–i).

Furthermore, despite the unstable surface deviating substantially from the reference, the norm of the velocity never grew unboundedly in this case. Thus determining the presence of instability based on the norm of the velocity alone may not accurately predict the presence of instabilities. The cause of this may be the significantly negative SMB which causes glacier thinning and in a sense "stabilizes" the solver over time. However, as is evident from Fig. 7b the solution is still polluted by the initial instability. For this reason, the presence of the vertical velocity shifting in sign spuriously was taken as a criteria for instability in this case.

Figure 7c shows that for the stable time-step size $\Delta t = 20$ yr, the error using FSSA is larger than without FSSA at the deep parts of the glacier, but the accuracy near edges is higher, so that the glacier area is more accurate (top left and lower right panel). This is contrary to what was found in the previous 2D experiment, where the FSSA method was overall more accurate. The reason for this discrepancy could be explained by the fact that the estimated reference solution in this case is only refined temporally and therefore may not represent the analytical solution accurately enough. Figure 7b and Fig. 7c also show that the less accurate the solution is, the smaller the glacier retreat and larger ice thickness compared to the reference solution. This should be contrasted with the first experiment which considered an advancing glacier, where the less accurate the solution, the less the glacier had advanced.

In order to ensure computation times are not negatively affected by the FSSA method, an experiment was performed to measure the CPU-time of the FSSA method compared to no stabilization. The computation times for different $\Delta t$ are shown in Table 5. It is seen that the FSSA method was faster for all cases. The FSSA method was about 10 % faster for the case with $\Delta t = 20$ yr, while the difference is only slightly in favor of FSSA for $\Delta t = 10$ yr and $\Delta t = 5$ yr. The difference in computation times seems to be related in this case to the average number of nonlinear iterations needed for convergence. For example, for the case $\Delta t = 20$ yr the FSSA method required about 10 % fewer iterations for convergence, which explains the 10 % difference in computation times. The increase in nonlinear iterations might be due to stability issues of the unstabilized solver as the relative difference becomes smaller with decreasing $\Delta t$.

In summary, the FSSA method gave a more stable solution, increasing the LST by at least a factor of two, without negatively
impacting computation times. In regards to accuracy, the FSSA method yielded larger ice thickness errors in the interior, while
the error was reduced at the glacier front. This has the implication that the FSSA method may be more suitable at predicting
future glacier extent, while the non-FSSA method is more accurate for determining future glacier thinning.

**Table 4.** Largest stable time-step size of a 200 year long simulation with and without FSSA stabilization for two different mesh resolutions
$\Delta x$.

| | Largest stable time-step size, Midtre Lovénbreen (yr) | |
|---|---|---|
| | Without FSSA $\theta = 0$ | With FSSA $\theta = 1$ |
| Coarse mesh $\Delta x \sim 50$ m | 20–25 | $\geq 50$ |
| Fine mesh $\Delta x \sim 25$ m | 20–25 | $\geq 50$ |

**Table 5.** CPU times for a 200 year long simulation for different time-step sizes $\Delta t$, measured with and without FSSA stabilization. The
simulations were run on a mesh with horizontal resolution $\Delta x \sim 25$ m. Note that case with $\Delta t = 40$ yr and $\theta = 0$ is unstable.

| | CPU time, Midtre Lovénbreen | |
|---|---|---|
| $\Delta t$ (yr) | Without FSSA $\theta = 0$ | With FSSA $\theta = 1$ |
| 5 | 4h 17m | 4h 14m |
| 10 | 2h 20m | 2h 16m |
| 20 | 1h 25m | 1h 15m |
| 40 | 1h 1m | 46m |

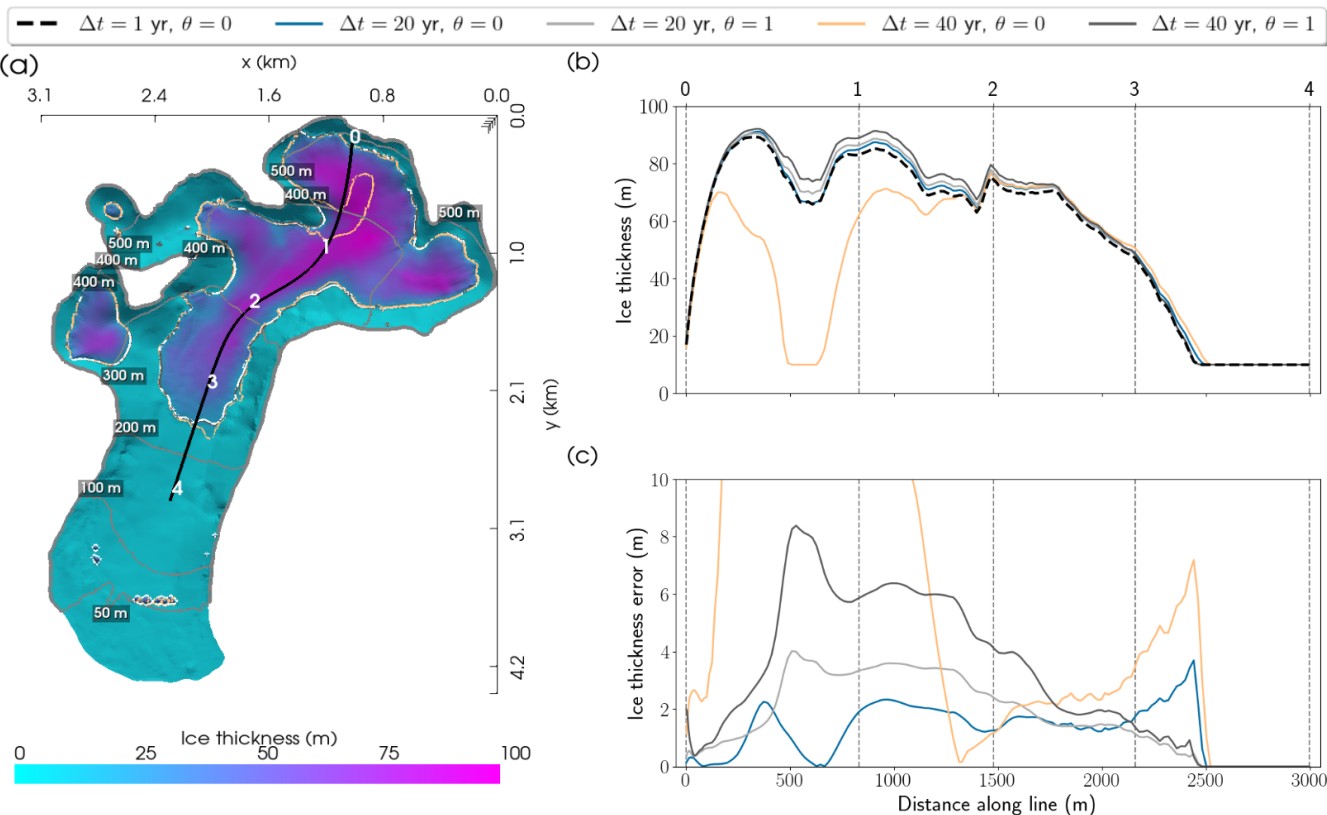

**Figure 7.** Midtre Lovénbreen at year 2195. The left panel (a) shows the glacier outlines of the reference solution (white) using a short time-step size $\Delta t = 1$ yr, as well as outlines for a simulation using a larger $\Delta t = 40$ yr, with FSSA (dark gray) and without FSSA (orange). The thickness of the glacier as given by the reference solution is indicated with colors in panel (a). The upper right panel (b) shows the ice thickness along the black line in panel (a), for the reference solution (dashed black line in (b)) and simulations with and without FSSA for $\Delta t = 20, 40$ yr (solid lines in (b)). The lower right panel (c) shows ice thickness errors as compared to the reference solution. The left panel (a) was created using the open source visualization and analysis toolkit PyVista (Sullivan and Kaszynski, 2019).

## 5 General discussion & conclusions

The FSSA was implemented into Elmer/Ice and tested on simulations of synthetic glaciers as well as on Midtre Lovénbreen,
Svalbard. The FSSA increased the largest stable time-step (LST) size by a factor of two for the simulation of Midtre Lovénbreen and up to a factor of five for the synthetic Perlin glacier with low surface slope. Low glacier surface slopes were correlated with a shorter LST for the unstabilized method and were also the cases where the stabilization had the greatest effect. This may be due to the thicker ice developed on the glaciers with low bedrock inclination, for which stability restrictions on lower order models have shown a strong inverse dependence (e.g., Gong et al., 2017; Robinson et al., 2022). For a case approaching
steady state, it was also found despite the large time-step sizes allowed for by the FSSA, the same steady state was approached compared to a reference simulation using a short time-step size. Even without stabilization the LST was already quite large in

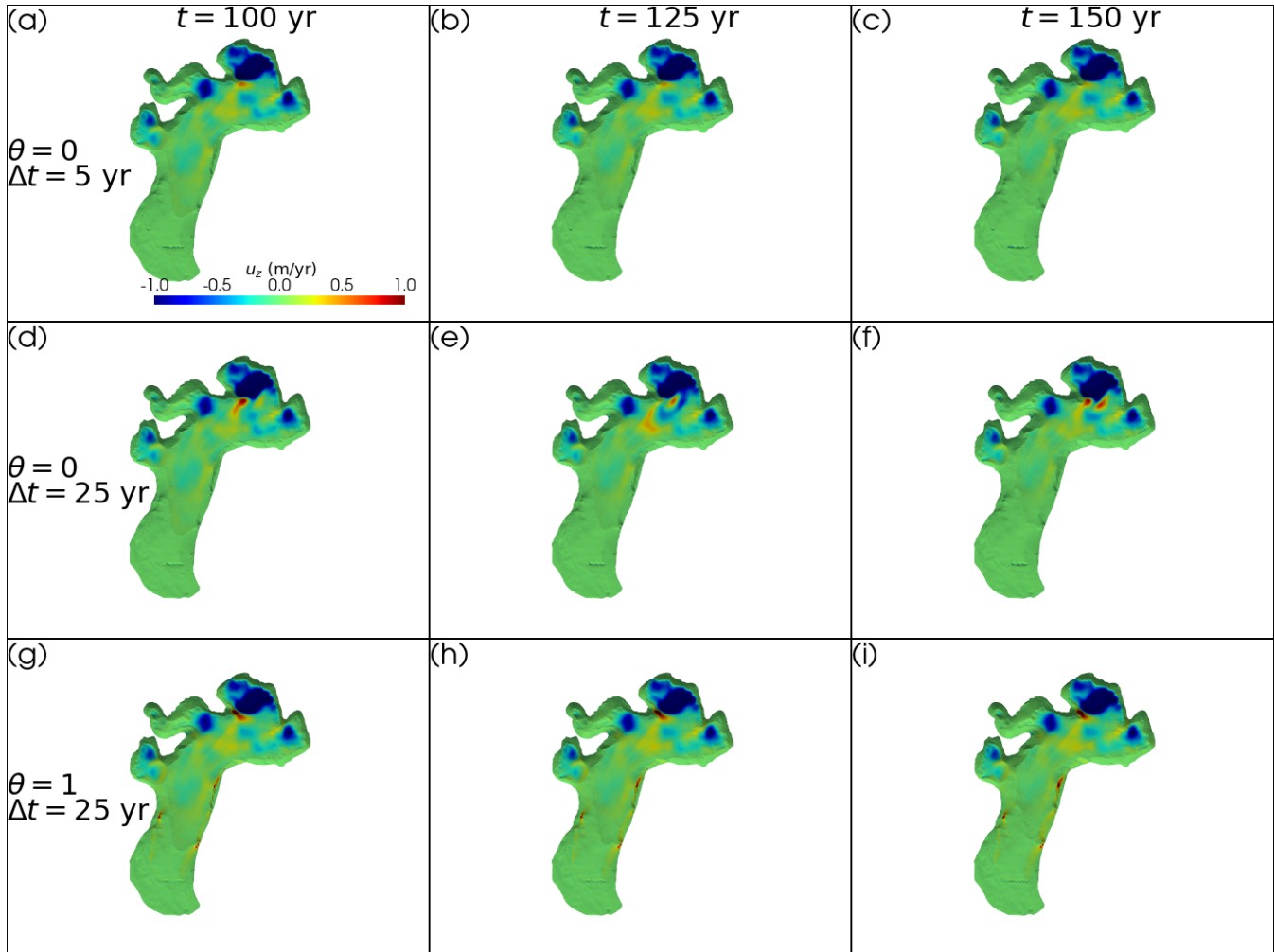

**Figure 8.** Vertical velocity profiles, $u_z$, at indicated times $t$ on Midtre Lovénbreen for cases (a-c) stable $\Delta t = 5$ yr without FSSA, (d-f) unstable $\Delta t = 25$ yr without FSSA and (g-h) stable $\Delta t = 25$ yr with FSSA. The figure was created using the open source visualization and analysis toolkit PyVista (Sullivan and Kaszynski, 2019).

many cases, for instance for the advancing Perlin glacier of intermediate slope a LST of 13 yr was observed, compared to 2 to 4 yr for the ice-sheet simulations in e.g., Löfgren et al. (2022). The larger LST observed here likely stem from a combination of lower flow speeds ($\sim 20$ m yr$^{-1}$ compared to $\sim 40$ m yr$^{-1}$) and the considerably thinner ice ($\sim 200$ m compared to $\sim 2000$ m).

It was also found that when compared to other means of stabilization, e.g., residual-free bubbles (RFB) and streamline upwind Petrov-Galerkin (SUPG), FSSA was the only one when used alone that increased the LST. However, combining FSSA and SUPG was the most stable choice. Furthermore, the mesh study revealed that the LST of the unstabilized case was mesh independent, in agreement with Löfgren et al. (2022), while the FSSA admitted a slight mesh dependence at high

spatial resolutions. However, even for the finest mesh resolution investigated the FSSA had twice the LST compared to no stabilization.

The FSSA mitigated instabilities and improved the accuracy overall, with the only exception being in the deep parts of Midtre Lovénbreen. As the computational cost is also low the FSSA can be added as a security measure in simulations to prevent sloshing instability polluting the result, without negatively impacting accuracy or simulation times. For glacier simulations this
may offer the greatest benefit of FSSA, given the already large time-step sizes of the unstabilized method. Another potential usage are spin-up simulations, where a very large time-step size of e.g., 50 yr could be used. Climate data could then be incorporated on shorter time scales using semi-implicit time stepping of the surface.

In the previous study by Löfgren et al. (2022) the FSSA was tested on synthetic ice-sheet experiments, largely disregarding the effect of variable bedrock and sliding conditions and simplifying the contact problem near glacier fronts. This paper
demonstrates that the FSSA method is applicable to more complex real-world simulations and the new implementation in Elmer/Ice makes the method accessible to a broad user base.

Some limitations of the current study are the generally low flow speeds ($< 100$ m yr$^{-1}$) and that only linear sliding laws were considered. It also remains to adapt the FSSA method to higher-order time-stepping schemes, which could possibly yield better stability properties due to stronger coupling between the geometry and velocity, as has been demonstrated by Wirbel and
445 Jarosch (2020) for a second-order Runge-Kutta scheme. Considering higher-order time stepping is for the authors an ongoing project.

*Code availability.* Elmer version `fededfbf7` (branch `devel`) has been used in this study and is downloadable from https://github.com/ ElmerCSC/elmerfem.

**Appendix A: Bedrock generation**

This section presents an algorithm that is used to random-generate bedrock topographies. It is based on a method that is common practice in computer graphics as a computationally inexpensive and flexible way of random generating visually appealing landscapes, clouds, textures etc., known as gradient noise. The first application of gradient noise, so-called Perlin noise, was developed by Perlin (1985) to model fire, water and wrinkled surfaces, and later adapted by Musgrave et al. (1989) for landscape generation. As the name suggests, gradient noise is based on generating a set of pseudo-random gradients at
predefined vertices and then interpolating polynomials matching these gradients, such that the resulting global interpolating function is smooth. In this study, the mesh vertices are assumed to be equally spaced with a spatial period $\Delta x^f$. The superscript $f$ is used to denote the fact that the final noise function may consist of multiple spatial frequencies, so-called *octaves* — similar to a Fourier decomposition. See Fig. A1 for an example of a random-generated bedrock consisting of three octaves.

For the interpolation cubic Hermite polynomials are used

$$p^k(x) = \sum_{i=0}^{3} c_i x^i, \tag{A1}$$

$$\tag{A2}$$

where $p^k$ is the polynomial interpolation in cell $k$, and $c_i$ are coefficients determined by matching gradients and function values at the vertices. This gives the set of equations for each vertex $\mathbf{x_i}$ in cell $k$

$$p^k(\mathbf{x}_i) = 0, \tag{A3}$$

$$\frac{dp^k(\mathbf{x}_i)}{dx} = f_{x_i}, \tag{A4}$$

where $f_{x_i}$ are uniformly random generated gradients over the interval $[-1, 1]$, corresponding to slope angles between $-45°$ and $45°$. Since the number of vertices in each cell are two, this leads to a total of four equations, matching the total number of unknowns. Solving the resulting linear system then gives the octave characterized by the spatial period $\Delta x^f$ of the noise function. The final noise function is obtained as a linear combination of all the octaves, such that

$$\text{noise}(x, y) = \sum_{f=1}^{n} A^f \text{octave}(\Delta x^f, x, y) \tag{A5}$$

where $A^f$ and $\Delta x^f$ are the amplitude and spatial period of the f:th octave, respectively, and $n$ is the number of octaves.

*Author contributions.* AL was responsible for conceptualization, project administration, investigation, methodology, software, validation, visualization, writing – original draft preparation. CH contributed with software, investigation and writing – review & editing. JA provided conceptualization, project administration, supervision, funding acquisition and writing – original draft preparation. PR and TZ both contributed with project administration, methodology, investigation, validation, software, resources, funding acquisition, writing – review & editing.

*Competing interests.* JA is a member of the editorial board of The Cryosphere. The peer-review process was guided by an independent editor, and the authors have also no other competing interests to declare.

*Acknowledgements.* Computations on CSC's platforms `mahti` and `c-pouta` were supported by the HPC-Europa3 program, part of the European Union's Horizon 2020 research and innovation program under grant agreement no. 730897. Josefin Ahlkrona have received funding from the Swedish Research Council, grant number 2021-04001, which funded Josefin Ahlkrona, Thomas Zwinger and André Löfgren. Thomas Zwinger's contribution was also under Academy of Finland COLD consortium grant no. 322978. Peter Råback has received funding

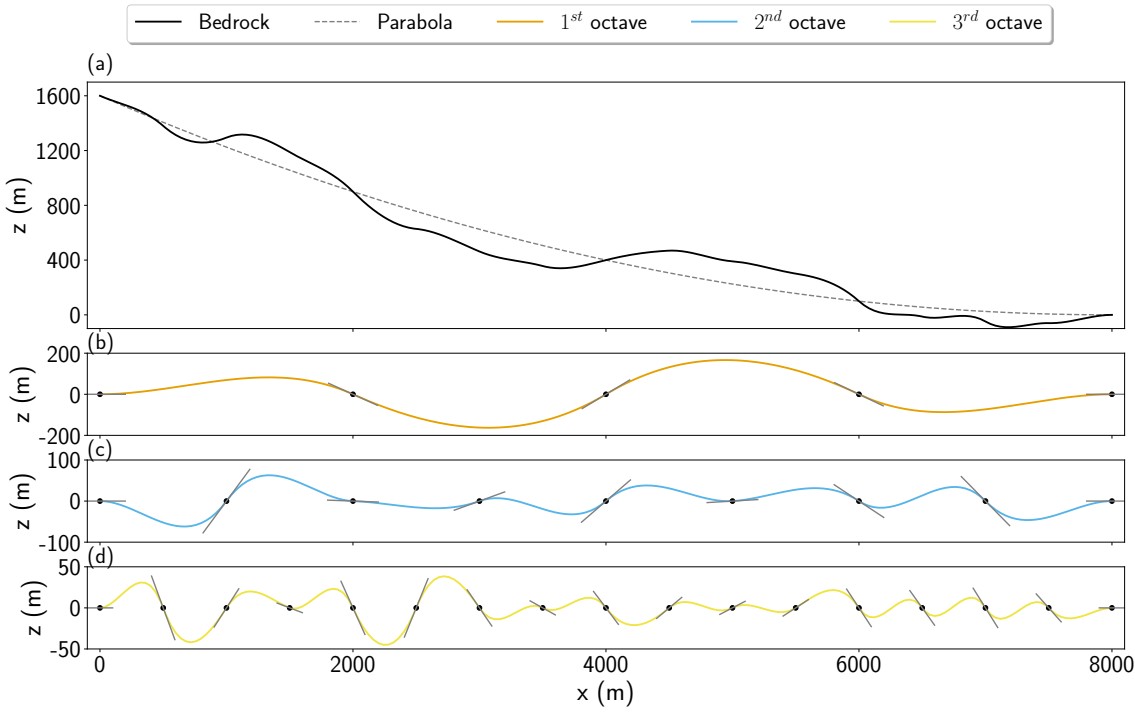

**Figure A1.** Gradient noise consisting of three octaves (orange, blue and yellow solid lines in (b–d)) superimposed on a parabola (gray dashed line in (a)) to generate a naturally looking bedrock topography (black solid line in (a)). Spatial periods of the octaves (b–d) are $\Delta x^1 = 2000$ m (orange solid line in (b)), $\Delta x^2 = 1000$ m (blue solid line in (c)) and $\Delta x^3 = 500$ m (yellow solid line in (d)). The black dots and gray solid lines in (b–d) denote the nodal values (zero in this case) and the matched pseudo-random generated tangents.

from the European High Performance Computing Joint Undertaking (JU) and Spain, Italy, Iceland, Germany, Norway, France, Finland and Croatia under grant agreement no. 101093038. Josefin Ahlkrona and Christian Helanow have received funding from Swedish e-Science Research Centre (SeRC).

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
