# Peer review of "Increasing numerical stability of mountain valley glacier simulations: implementation and testing of free-surface stabilization in Elmer/Ice"

_EGUsphere, 2023_

## Referee Comment (RC1)

**Review of 'Increasing numerical stability of mountain valley glacier simulations: implementation and testing of free-surface stabilization in Elmer/Ice'**

September 2023

**1 Summary**

In this manuscript the authors describe a technique for approximating the Backward Euler method for time stepping in a coupled Stokes-free surface system for glacier evolution. The so-called free surface stabilization algorithm only involves the inclusion of an additional term in the Stokes equations, and so is cheap to implement and seems to increase the size of allowable time steps (though probably not to the same degree that a proper implicit solver would). The main difference between this paper and a previous one by the same author on this subject is its implementation in Elmer/Ice and its application to a realistic geometry.

I think the paper is a nice contribution and the method described is potentially useful. Unfortunately, I think that the work suffers from a lack of specificity that hampers a careful reader from really understanding the performance characteristics of the method. Towards the purpose of improving clarity and providing a more sober view of what can be expected of this method, I have included some comments below.

**2 Line-by-line comments**

**L29** I don't think models 'suffer' from time-step restrictions, but they are subject to them. They are not necessarily 'parabolic' either - when flow is dominated by bedrock slopes, the equations have a more hyperbolic character.

**L32** Here and elsewhere, the word 'stability' is used without precision. How is this concept characterized here? Is it just the lack of visibily detectable wiggles? Is it when a simulation blows up?

**L120** This comparison is a bit contrived because it relies on one particular paradigm for solving free-surface Stokes, namely that the nonlinear coupling is managed via Picard iteration between the velocity and thickness solves. There are alternatives: solving simultaneously with Picard, using Newton's method (although these are admittedly both easier to implement in a terrain-following coordinate system). These alternatives don't necessarily involve solving Stokes more than once in the way that is described here. It's important to be specific!

**Eq. 3.1** This might be an appropriate place (although there are others) to mention the very important condition for all of your equations, namely that they are only valid when $z_s > z_b$! Also, how do you deal with this constraint (presuming that you do, because the 'ice-free' region in the Midtre Lovenbreen experiment expands).

**L132** These 'appropriate function spaces' are never stated explicitly. Presumably the Taylor-Hood element is used here?

**Eq. 11** How is the transport equation discretized in space? A finite element method? If so, which function space? If it's solved nodally, then how is the spatial derivative in surface elevation calculated? This is an advection equation, so often requires a stabilization scheme, e.g. upwinding. Is that done here? Does whatever representation of the surface elevation satisfy an inf-sup condition?

**L150** Is this supposed to be referencing Eq. 12? If so should it be that the first term on the *right* side of Eq. 12 is zero?

**Eq. 13** Maybe worthwhile to say that you're using a forward Euler discretization of the time-derivative in Eq. 12. Also, the superscript on $\Omega$ seems to be messed up.

**Eq. 14** Does the $u$ that appears in the 'new' part of the weak form need a superscript too?

**Eq. 15** I'm not sure that including the equation adds anything here. It might be clearer to just write that 'in the case of the SIA, the FSSA coincides precisely with evaluating the pressure at the end of the time integration. In the case of the Stokes' equations, this is an approximation, etc.'

**Eq. 18** It's worth noting that with this strictly non-negative mass balance and no way for mass to enter or leave the system, that this glacier will grow without bound. This is unfortunate because it would be interesting to see the result of applying this method to grow a glacier to steady state.

**L210** I think it's unfortunate that the solution is only compared to other model results run with a smaller time step. A more robust and complementary approach would be to evaluate this method against a manufactured solution. This would also potentially provide insight into the ambiguous

results later about whether the FSSA is more or less accurate than without.

**L216** Again, how is stability defined? Is the LST computed by using bigger and bigger time steps until the solver produces NaNs?

**L240** Could this be explained in a way that relates more closely to theory? In principle, so long as the CFL condition is satisfied, the forward Euler and backward Euler (which the FSSA approximates) have the same order of numerical accuracy. Why would the accuracy deviate between how the time derivative is discretized?

**Fig. 3** I really struggle to distinguish between the lines. Can these be made thicker, or the plot larger or something to make this more easily seen?

**L302** What is the 'derivative of the viscosity'? Do you mean the whole Jacobian? If so, then including a relaxation parameter is pretty standard.

**L324** I don't understand the notion of higher or lower accuracy for advancing or retreating glaciers. This needs to be justified or removed.

**L329** Did these instabilities only appear in the absence of the FSSA or with it too? Later text seems to indicate the former, but it's not clear here. What does it mean for an instability to be 'specific to the setup'? That would seem like a very bad property to not know whether a simulation is going to be stable or not a priori. I can't see how 'other' instabilities are being suppressed here - this doesn't seem to be shown.

**Table 2** How do we know its 20 and not, say 17.8 or something?

**L355** I don't understand what a 'more viscous behavior' means.

**Appendix A** I don't think it's all that relevant as to how bedrock was generated (there are many methods of doing this, e.g. Gaussian random fields, random fourier features, etc.), but nonetheless this section is quite opaque. It might be better just to reference something rather than include this sort of insufficient description.

---

## Referee Comment (RC2)

The manuscript "Increasing numerical stability of mountain valley glacier simulations: implementation and testing of free-surface stabilization in Elmer/Ice" applies a method developed in the context of mantle-convection modeling to stabilize the evolution of the free surface. The method has been already studied in the context of ice sheet modeling but with simplified problems. I think the paper can be useful to the ice sheet community. However, the presentation should be improved and critical details about the discretization of the equations should be added. In general, I wish the authors were more ambitious with this work, trying to understand better the method from a theoretical point of view, considering higher-order time-discretizations and exploring the effect of different spatial discretizations of the free-surface equation with the coupling method proposed. Below are more detailed comments.

Eq(7): This is a pet peeve of mine, but I think that calling the friction coefficient $\beta^2$ is really bad notation despite being commonly used in ice sheet modeling. It makes you think that its square root is a meaningful physical quantity. Why not just call it $\beta$, or $C$ or $\mu$?

Section 3.1: This section contrasts an explicit solution of the Stokes free-surface equation with an implicit solution of such equations. However, there are hidden assumptions in both approaches. As for the explicit solution, it seems that the authors have in mind a low-order, explicit time-integration scheme like Forward Euler. However, other high-order scheme could be used. As an example, one could use the Runge-Kutta scheme, where the Stokes equation would need to be solved multiple time per time step (at each stage of the scheme). Regarding the implicit solution, in addition of limiting the analysis to a low-order scheme like Backward Euler, they also assume that the coupled Stokes free-surface problem would be solved by iterating the solution of Stokes and the free-surface problem until convergence. This implies an outer nonlinear iteration loop (at each iteration the Stokes and free-surface equations are solved) and an inner nonlinear iteration loop for solving the Stokes problem. While this might be the simplest method to implement, it is likely not the most efficient. In fact, one could consider the Stokes and free-surface coupled problem in a monolithic fashion and solve it with a single iterative scheme, linearizing at the same time Stokes and the coupled Stokes free-surface problem. I recommend that the authors better explain their choices and possible alternatives.

Eq (11): I would write here eq (10) here as well, with $u$ evaluated at times $t^k$ and all the domains evaluated at time $t^k$ except for the forcing term that is evaluated at time $t^{k+1}$. This is the scheme you are after, that is, account for the evaluation of the forcing term at the time $t^{k+1}$ to anticipate the effect of the domain change at least on the forcing term.

Figure 2: Please mention at least in the caption that these are not the only two time-stepping / coupling options.

Line 131: Definition of inner product. Inner product has two arguments. What you wrote and what you used in eq. (10) is just an integral over the domain $\Omega$. If you want to call it inner product then use the notation $(\cdot, \cdot)_\Omega$.

Lines 150-153: This part is confusing. I think there are a couple of typos. Eq (13) is referenced instead of equation (12), and "first term on the left-had side" should be "first term on the right-hand side". Further, it is not true that in eq. (12) the domain is assumed to move only due to the velocity of the deformation. Eq. (12) is Reynolds theorem and is valid in general. However, $u_b$ is the velocity of the ice boundary, which is not the velocity of the ice at the boundary. At the surface, $u_b = u + a_s \hat{z}$, that is, the velocity of the surface is the sum of the velocity of the ice $u$ and of the accumulation/ablation rate. Please rephrase this paragraph.

Eq (13): typo, check the subscripts.

Section 3: Please add details about the discretization of the Stokes equations and the free surface equation. In particular, are you using any stabilization for (11), e.g., upwind, flux limiters, SUPG? How the spatial discretization of (11) and (14) affect the effectiveness of the proposed approach?

Section 4.2.1: In addition to the results presented, it would be informative to have results where the same (relatively fine) mesh is used for all the simulation (including the reference one). This would help separating the effect of the spatial discretization from the time discretization, which is the main focus of the paper.

Appendix A: Can you detail how the "random generated gradients" are generated?

---

## Author Response (AR1)

Dear Dr. Johannes Fürst and Referees,

We once again thank you for the work you have put in to providing us with highly valuable feedback, which has now been included in the manuscript. Please find below our edited point-by-point response to your comments. Our previous responses are found under the paragraphs labeled **RESPONSE** and our edited responses under **EDIT**. All line numbers refer to the marked-up version of the manuscript.

Sincerely,

André Löfgren, Josefin Ahlkrona, Thomas Zwinger, Peter Råback, and Christian Helanow

**Reviwer One Comments**

**COMMENT**: In this manuscript the authors describe a technique for approximating the Backward Euler method for time stepping in a coupled Stokes-free surface system for glacier evolution. The so-called free surface stabilization algorithm only involves the inclusion of an additional term in the Stokes equations, and so is cheap to implement and seems to increase the size of allowable time steps (though probably not to the same degree that a proper implicit solver would). The main difference between this paper and a previous one by the same author on this subject is its implementation in Elmer/Ice and its application to a realistic geometry. I think the paper is a nice contribution and the method described is potentially useful. Unfortunately, I think that the work suffers from a lack of specificity that hampers a careful reader from really understanding the performance characteristics of the method. Towards the purpose of improving clarity and providing a more sober view of what can be expected of this method, I have included some comments below.

**RESPONSE**: Thank you for this nice summary, and for your suggestions for improving the clarity of the manuscript. Below you will find our point-by-point response to each of your comments.

**Comments**

**COMMENT 1**: L29: I don't think models 'suffer' from time-step restrictions, but they are subject to them. They are not necessarily 'parabolic' either - when flow is dominated by bedrock slopes, the equations have a more hyperbolic character.
**RESPONSE**: We'll change the wording, and specify that the parabolic time-step size constraint is valid under shear-dominated flow.
**EDIT**: See lines 29–30.

**COMMENT 2**: L32: Here and elsewhere, the word 'stability' is used without precision. How is this concept characterized here? Is it just the lack of visibly detectable wiggles? Is it when a simulation blows up?
**RESPONSE**: By instability we mean the usual magnification of truncation and round-off errors. We'll define what we mean by instability in the introduction and for each experiment add a sentence on how we detect them. However, finding a consistent criteria for detecting instability has proven to be difficult in this study. For example in the Perlin case, detecting instability by means of checking the norm of the vertical velocity was useful to determine the presence of instability. On the other hand, for the Midtre-Lovénbreen case, the solution didn't blow up despite being considerably off from the reference, as indicated by Fig. 1 below (Fig. 8 in the manuscript). We believe this is due to the considerably negative surface-mass balance in that case, which causes glacier thinning,

and in a sense "stabilizes" the solver such that it never blows up, but is still polluted by the initial instability. For this reason we in this case take the presence of sloshing (i.e., spurious shifts in the sign of the vertical velocity) to indicate instability, rather than the norm of the vertical velocity blowing up to infinity. We'll elaborate on this point in the manuscript.

**EDIT**: Stability is now defined in the introduction at line 35, and the criteria for instability in the Perlin case is described at line 246–249 and for Midtre Lovénbreen at lines 386–388.

[Figure]

Figure 1: Midtre Lovénbreen at year 2195. The left panel (a) shows the glacier outlines of the reference solution (white) using a fine time-step size $\Delta t = 1$ yr, as well as outlines for a simulation using a larger $\Delta t = 40$ yr, with FSSA (dark gray) and without FSSA (orange). The thickness of the glacier as given by the reference solution is indicated with colors in panel (a). The upper right panel (b) shows the ice thickness along the black line in panel (a), for the reference solution (dashed black line in (b)) and simulations with and without FSSA for $\Delta t = 20, 40$ yr (solid lines in (b)). The lower right panel (c) shows ice thickness errors as compared to the reference solution.

**COMMENT 3**: L120: This comparison is a bit contrived because it relies on one particular paradigm for solving free-surface Stokes, namely that the nonlinear coupling is managed via Picard iteration between the velocity and thickness solves. There are alternatives: solving simultaneously with Picard, using Newton's method (although these are admittedly both easier to implement in a terrain-following coordinate system). These alternatives don't necessarily involve solving Stokes more than once in the way that is described here. It's important to be specific!

**RESPONSE**: We'll clarify that this specifically corresponds to a Picard linearization.

**EDIT**: See the paragraph at lines 127–130.

**COMMENT 4**: Eq. 3.1: This might be an appropriate place (although there are others) to mention the very important condition for all of your equations, namely that they are only valid when $z_s > z_b$! Also, how do you deal with this constraint (presuming that you do, because the 'ice-free' region in the Midtre Lovénbreen experiment expands).

**RESPONSE**: Thank you, we'll add a paragraph elaborating on this.

**EDIT**: See lines 112–116.

**COMMENT 5**: L132: These 'appropriate function spaces' are never stated explicitly. Presumably the Taylor-Hood element is used here?

**RESPONSE**: Appropriate function spaces refers to those satisfying the so-called *inf-sup* condition, e.g., Taylor-Hood elements. In case the function spaces do not satisfy this condition, as is the case for equal-order bilinear element (e.g., P1-P1 elements), stabilization has to be introduced into the weak formulation in order to circumvent this condition. In our study we are using the P2-P1 Taylor-Hood elements for 2D, while due to the 3D case being much more computationally expensive we opt to use a GLS stabilized formulation with P1-P1 elements. We'll insert a sentence in the methodology for each experiment where we mention this.

**EDIT**: To clarify, we have for each experiment added a sentence specifying what element we have used. Relevant lines are 251–252 and lines 380–382 for 2D and 3D, respectively.

**COMMENT 6**: Eq. 11: How is the transport equation discretized in space? A finite element method? If so, which function space? If it's solved nodally, then how is the spatial derivative in surface elevation calculated? This is an advection equation, so often requires a stabilization scheme, e.g. upwinding. Is that done here? Does whatever representation of the surface elevation satisfy an inf-sup condition?

**RESPONSE**: Yes, it's discretized using FEM, and we do indeed use upwinding, specifically we're using residual-free bubbles for the 2D case and SUPG in 3D. We'll add a section (Sect. 3.4) on the spatial discretization and stabilization of the free-surface equation. Regarding the inf-sup condition, we're not sure what you are referring to; do you mean an inf-sup stability restriction between the surface $h$ and the velocity $\mathbf{u}$ as when they are treated as unknowns in a monolithic fashion? Interesting to think about, but we do not take any such restriction into account.

**EDIT**: The spatial discretization of the free-surface equation is described in section 3.4, see lines 190–196. Regarding your comment about the inf-sup condition, since the explicit time stepping approach taken here decouples the Stokes

equation from the free-surface equation, it is not a min-max problem and as such not subject to an inf-sup condition.

**COMMENT 7**: L150: Is this supposed to be referencing Eq. 12? If so should it be that the first term on the right side of Eq. 12 is zero?
**RESPONSE**: Thank you for catching this. You are indeed right.
**EDIT**: We have now fixed this typo.

**COMMENT 8**: Eq. 13: Maybe worthwhile to say that you're using a forward Euler discretization of the time-derivative in Eq. 12. Also, the superscript on $\Omega$ seems to be messed up.
**RESPONSE**: Thank you. We'll fix the superscript and add a sentence mentioning that we are using forward Euler.
**EDIT**: Mentioned in line 171.

**COMMENT 9**: Eq. 14: Does the $u$ that appears in the 'new' part of the weak form need a superscript too?
**RESPONSE**: Good catch, it should indeed.
**EDIT**: Fixed.

**COMMENT 10**: Eq. 15: I'm not sure that including the equation adds anything here. It might be clearer to just write that 'in the case of the SIA, the FSSA coincides precisely with evaluating the pressure at the end of the time integration. In the case of the Stokes' equations, this is an approximation, etc.
**RESPONSE**: Very elegantly put, we'll adopt this in the manuscript.
**EDIT**: See the paragraph at lines 184–186.

**COMMENT 11**: Eq. 18: It's worth noting that with this strictly non-negative mass balance and no way for mass to enter or leave the system, that this glacier will grow without bound. This is unfortunate because it would be interesting to see the result of applying this method to grow a glacier to steady state
**RESPONSE**: Thank you for the suggestion. We'll add such an experiment for two-dimensional Perlin case. The experiment we have in mind is to start from the surfaces obtained at the end of the current simulations, and then introduce ablation into the SMB and continue the simulation for a few hundred years (until a steady state is reached).
**EDIT**: The experiment is described in the new section 4.2.2 (lines 255–275), and the results in section 4.2.4 (lines 340–355).

**COMMENT 12**: L210: I think it's unfortunate that the solution is only compared to other model results run with a smaller time step. A more robust and complementary approach would be to evaluate this method against a manufactured solution. This would also potentially provide insight into the ambiguous results later about whether the FSSA is more or less accurate than without.
**RESPONSE**: Thank you for the suggestion, comparing against a manufactured solution would indeed be interesting. However, it is our experience that such solutions are far from real-world applications; constructing a manufactured solution that represent such cases we think is beyond the scope of this paper.

**COMMENT 13**: L216: Again, how is stability defined? Is the LST computed by using bigger and bigger time steps until the solver produces NaNs?
**RESPONSE**: Please see our answer to comment 2.

**COMMENT 14**: L240: Could this be explained in a way that relates more closely to theory? In principle, so long as the CFL condition is satisfied, the forward Euler and backward Euler (which the FSSA approximates) have the same order of numerical accuracy. Why would the accuracy deviate between how the time derivative is discretized?
**RESPONSE**: Good point, which we'll elaborate on. We should indeed expect this, but since the method is just an approximation of backward Euler, we thought it would be a good sanity check to confirm that the order of accuracy for stable solution is what we expect, i.e., linear.
**EDIT**: We mention at line 313 that this is expected as long as stability conditions are satisfied.

**COMMENT 15**: Fig. 3: I really struggle to distinguish between the lines. Can these be made thicker, or the plot larger or something to make this more easily seen?
**RESPONSE**: We'll make the lines thicker and the plots bigger.
**EDIT**: Thank you for this suggestion. We made the velocity plot smaller, increased the thickness of dashed lines slightly and cropped the image to remove as much white space as possible.

**COMMENT 16**: L302: What is the 'derivative of the viscosity' ? Do you mean the whole Jacobian? If so, then including a relaxation parameter is pretty standard.
**RESPONSE**: That is indeed the case. The "new" refers to new in Elmer/Ice, we'll clarify this.
**EDIT**: Since you pointed out that it is a standard method, we decided to simply mention that we use relaxation parameter and its value.

**COMMENT 17**: L324: I don't understand the notion of higher or lower accuracy for advancing or retreating glaciers. This needs to be justified or removed.
**RESPONSE**: We'll remove this sentence.
**EDIT**: Removed at lines 419-420.

**COMMENT 18**: L329: Did these instabilities only appear in the absence of the FSSA or withit too? Later text seems to indicate the former, but it's not clear here. What does it mean for an instability to be 'specific to the setup' ? That would seem like a very bad property to not know whether a simulation is

going to be stable or not a priori. I can't see how 'other' instabilities are being suppressed here - this doesn't seem to be shown.

**RESPONSE**: For the cases presented, no such instabilities occurred for FSSA. But we have during the review process found cases where the spikes also occur for FSSA, so we'll remove this conclusion.

**EDIT**: Removed at lines 424–429.

**COMMENT 19**: Table 2: How do we know its 20 and not, say 17.8 or something?

**RESPONSE**: We tested also for $\Delta t = 25$ yr, so you're right it is something between 20 yr and 25 yr. However, we report 20 yr as the LST since it was the largest time-step size we tested and had a stable solver. This does not affect our conclusion that the FSSA allows for at least twice as large time steps. We'll clarify this in the manuscript.

**EDIT**: The reported LST is specified in the range 20-25 yr in Table 4.

**COMMENT 20**: L355: I don't understand what a 'more viscous behavior' means.

**RESPONSE**: By more viscous behavior we mean a larger shear- to slip-velocity ratio, which we'll clarify in the manuscript.

**EDIT**: We removed this formulation and instead connect the smaller LST to the thicker ice developed on the glacier with smaller bedrock inclination. See lines 449–452.

**COMMENT 21**: Appendix A: I don't think it's all that relevant as to how bedrock was generated (there are many methods of doing this, e.g. Gaussian random fields, random fourier features, etc.), but nonetheless this section is quite opaque. It might be better just to reference something rather than include this sort of insufficient description.

**RESPONSE**: It is indeed not the focus of the study, but we felt that it was warranted to include as it gives the reader insight into how we constructed our glacier. We'll, however, shorten it down to make it more concise.

**EDIT**: We removed the description of the algorithm for the 3D case.

**Reviewer Two Comments**

**COMMENT**: The manuscript "Increasing numerical stability of mountain valley glacier simulations: implementation and testing of free-surface stabilization in Elmer/Ice" applies a method developed in the context of mantle-convection modeling to stabilize the evolution of the free surface. The method has been already studied in the context of ice sheet modeling but with simplified problems. I think the paper can be useful to the ice sheet community. However, the presentation should be improved and critical details about the discretization of the equations should be added. In general, I wish the authors were more ambitious with this work, trying to understand better the method from a theoretical point of view, considering higher-order time-discretizations and exploring the effect of different spatial discretizations of the free-surface equation with the coupling method proposed.

**RESPONSE**: Thank you for sharing your overall impression, it pleases us that you believe our paper might be useful to the ice-sheet community. To raise our ambition to your expectations we've decided to include two more studies. In the first study we consider a retreating glacier starting from the final surface obtained in the Perlin glacier case; within this experiment we'll explore how upwinding (residual-free bubbles) affects the LST (largest stable time-step size). In the second experiment we'll include multiple mesh resolutions to study the effect of mesh resolution on the LST. However, while we also believe that considering higher order time-stepping is of great interest we have, given what we believe is feasible within this review, decided to limit the current study to only look at first order time stepping. We'll mention, however, that this is something to consider for future studies. It's also something we're already planning to do within a future project.

**Comments**

**COMMENT 1**: Eq. (7): This is a pet peeve of mine, but I think that calling the friction coefficient $\beta^2$ is really bad notation despite being commonly used in ice sheet modeling. It makes you think that its square root is a meaningful physical quantity. Why not just call it $\beta$, or $C$ or $\mu$?
**RESPONSE**: The $\beta^2$, I believe, is to indicate that it is a positive quantity. We'll follow your suggestion instead and use $\beta$ to denote the friction coefficient and explicitly state that $\beta \geq 0$.
**EDIT**: We've now changed the notation.

**COMMENT 2**: Section 3.1: This section contrasts an explicit solution of the Stokes free-surface equation with an implicit solution of such equations. However, there are hidden assumptions in both approaches. As for the explicit solution, it seems that the authors have in mind a low-order, explicit timeintegration scheme like Forward Euler. However, other high-order scheme could be used. As an example, one could use the Runge-Kutta scheme, where the Stokes equation would need to be solved multiple time per time step (at each stage of the scheme). Regarding the implicit solution, in addition of limiting the analysis to a low-order scheme like Backward Euler, they also assume that the coupled Stokes free-surface problem would be solved by iterating the solution of Stokes and the free-surface problem until convergence. This implies an outer nonlinear iteration loop (at each iteration the Stokes and free-surface equations are solved) and an inner nonlinear iteration loop for solving the Stokes problem. While this might be the simplest method to implement, it is likely not the most efficient. In fact, one could consider the Stokes and free-surface coupled problem in a monolithic fashion and solve it with a single iterative scheme, linearizing at the same time Stokes and the coupled Stokes free-surface problem. I recommend that the authors better explain their choices and possible alternatives.

**RESPONSE**: You're right that also higher-order explicit methods such as RK4 would involve solving the Stokes equations multiple times in each time step. We'll clarify in the manuscript that we're considering first-order time stepping. Regarding the implicit solver, other linearizations than the Picard linearization we have in mind might be more efficient; the point, however, is that implicit time stepping in general is considerably more expensive than explicit. Furthermore we believe it's a relevant example as it is the only implicit scheme we are aware of implemented in a large ice-sheet solver (e.g., Elmer/Ice). We'll also mention this in the manuscript.

**EDIT**: Thank you for pointing this out, we now mention that the explicit time-stepping scheme is first order in line 119, and also clarify that the implicit scheme is an example of a first order scheme available in Elmer/Ice in line 129.

**COMMENT 3**: Eq. (11): I would write here eq (10) here as well, with u evaluated at times $t^k$ and all the domains evaluated at time $t^k$ except for the forcing term that is evaluated at time $t^{k+1}$. This is the scheme you are after, that is, account for the evaluation of the forcing term at the time $t^{k+1}$ to anticipate the effect of the domain change at least on the forcing term.

**RESPONSE**: Thank you, we'll rephrase this.

**EDIT**: We reformulated most of section 3.3 to accommodate this change.

**COMMENT 4**: Figure 2: Please mention at least in the caption that these are not the only two time-stepping / coupling options.

**RESPONSE**: We'll clearly specify that this particular example, as used in Elmer, corresponds to a Picard linearization.

**EDIT**: We now mention in the caption that this is just an example.

**COMMENT 5**: Line 131: Definition of inner product. Inner product has two arguments. What you wrote and what you used in eq. (10) is just an integral over the domain $\Omega$. If you want to call it inner product then use the notation $(\cdot, \cdot)_\Omega$.

**RESPONSE**: Thank you, we'll change the notation.

**EDIT**: We have clarified that the inner product refers to the Frobenius notation, but otherwise decided to keep the notation as is. See lines 140–141.

**COMMENT 6**: Lines 150-153: This part is confusing. I think there are a couple of typos. Eq (13) is referenced instead of equation (12), and "first term on the left-had side" should be "first term on the right-hand side". Further, it is not true that in eq. (12) the domain is assumed to move only due to the velocity of the deformation. Eq. (12) is Reynolds theorem and is valid in general. However, $u_b$ is the velocity of the ice boundary, which is not the velocity of the ice at the boundary. At the surface, $u_b = u + a_s \hat{z}$, that is, the velocity of the surface is the sum of the velocity of the ice $u$ and of the accumulation/ablation rate. Please rephrase this paragraph.
**RESPONSE**: We'll fix the typos and rephrase this paragraph.
**EDIT**: The whole paragraph has been rephrased, please see lines 159–186.

**COMMENT 7**: Eq. (13): typo, check the subscripts.
**RESPONSE**: Thank you for catching this.
**EDIT**: Fixed.

**COMMENT 8**: Section 3: Please add details about the discretization of the Stokes equations and the free surface equation. In particular, are you using any stabilization for (11), e.g., upwind, flux limiters, SUPG? How the spatial discretization of (11) and (14) affect the effectiveness of the proposed approach?
**RESPONSE**: Thank you, we'll add a paragraph on the spatial discretization of Stokes in Sect. 3.2. In particular we'll state the elements and stabilization used. For the free-surface equation we'll add another subsection on the weak formulation and state the stabilization scheme we're using (residual-free bubbles for Perlin and SUPG for Midtre Lovénbreen). To address your concern regarding how the spatial discretization affect the FSSA, we'll include a 2D experiment evaluating the impact of adding upwinding into Eq. (11). For this purpose we'll evaluate the largest stable time-step size (LST) for the three cases: FSSA and upwinding, FSSA and no upwinding, no FSSA and no upwinding.
**EDIT**: We've clarified in section 3.2 (lines 141–143) what the appropriate function spaces are, and then we mention for each experiment the elements used; in this case Taylor-Hood for 2D (lines 250–252) and P1-P1 elements with GLS stabilized formulation for the Midtre Lovénbreen case (lines 380–382). The weak formulation of the free-surface equation is described in section 3.4. The setup of the new experiment is described in section 4.2.2 and the results in section 4.2.4. For this experiment we also considered SUPG, in addition to residual-free bubbles.

**COMMENT 9**: Section 4.2.1: In addition to the results presented, it would be informative to have results where the same (relatively fine) mesh is used for all the simulation (including the reference one). This would help separating the effect of the spatial discretization from the time discretization, which is the main focus of the paper

**RESPONSE**: Good suggestion, we'll add such a case in the experiment considering different mesh resolutions.

**EDIT**: We decided to simply use the fine mesh in all cases as you suggested, and made it a separate experiment to investigate the effect of the spatial resolution on the LST. The latter experiment is summarized in Table 2.

**COMMENT 10**: Appendix A: Can you detail how the "random generated gradients" are generated?

**RESPONSE**: The gradients are random generated from a uniform distribution over [-1, 1], i.e., gradients with slope angles between $-45°$ and $45°$. We'll mention this in the manuscript.

**EDIT**: This is now described in the Appendix at lines 500–501.

**List of relevant changes**

Below is a list major changes made to the manuscript.

1. Changed author ordering to conform to the ordering of the affiliations.

2. **Lines 4–6**: Adapted abstract to the LST obtained using a finer mesh on the advancing Perlin case.

3. **Lines 6–7**: Adapted abstract to the experiment comparing FSSA to transport stabilizations.

4. **Lines 38–39**: Defined instability (in)stability in Introduction

5. **Lines 58–59**: Highlighted difference between Löfgren et al. (2022) and this study.

6. Changed $\beta^2$ to $\beta$ and specified $\beta > 0$.

7. **Lines 112–116**: Described the minimum ice thickness constraint for the free-surface equation.

8. **Line 127**: Clarified that the implicit scheme corresponds to a Picard linearization.

9. **Line 129**: Mention that the explicit scheme in Fig. 2b is a first order scheme.

10. **Line 140**: Clarified that the inner product refers only to the colon operator.

11. **Lines 141–143**: Added a sentence stating what appropriate function spaces refers to.

12. **Lines 145–146**: Removed "new" part.

13. **Section 3.3**: Reformulated the whole section.

14. **Section 3.4**: Added section for the spatial discretization of the free surface.

15. **Line 203**: Added description of new experiments into the overview.

16. **Section 4.2**: Divided the Perlin setup into sections 4.2.1 and 4.2.2 for the advancing and retreating case respectively.

17. **Line 245**: Added a paragraph describing the setup of the mesh study.

18. **Line 251**: State the elements used for the Perlin experiment.

19. **Section 4.2.2**: Added new section describing the setup of the retreating glacier case.

20. **Lines 277–283**: Adapted reported stable time-step sizes to the finer mesh used.

21. **Line 284–288**: Added paragraph in the results section for the mesh study.

22. **Section 4.2.4**: Added results section for the retreating Perlin glacier experiment.

23. **Table 1**: Modified values in table for the finer mesh (1000, 10)

24. **Table 2**: Added new table comparing the LST for different mesh sizes and stabilization parameters.

25. **Table 2**: Added new table comparing the LST for different upwinding schemes and stabilization parameters.

26. **Figure 2b**: Replaced initial Stokes solve with an initial guess.

27. **Figure 3**: Regenerated plots for the finer mesh (1000, 10), made dashed lines thicker, velocity plots smaller and cropped the image.

28. **Figure 4**: Regenerated plots for the finer mesh (1000, 10)

29. **Figure 5**: Added figuring showing initial bedrock and surface velocities for the retreating Perlin case.

30. **Line 380**: Added statement for the Midtre Lovénbreen case which elements we use, and that we are using a GLS stabilized formulation.

31. **Line 386**: Added statement on how we detect instabilities on Midtre Lovénbreen.

32. **Lines 408–413**: Added paragraph on the issue of detecting instability by only considering the norm of the velocity on Midtre Lovénbreen.

33. **Lines 423–428**: Removed paragraph on spikes appearing in deglaciated areas on Midtre Lovénbreen.

34. **Table 4**: Specify LST in range 20–25 yr.

35. **Figure 8**: Added figure showing the appearance of sloshing on Midtre Lovénbreen.

36. Relabeled section "Conclusions" to "General discussion & conclusions".

37. **Lines 451–455**: Added discussion on results from the retreating Perlin experiment.

38. **Line 455**: Quantified differences in LST between Löfgren et al. (2022) and this study.

39. **Lines 451–455**: Added discussion on the experiment comparing FSSA to transport stabilization.

40. **Lines 475–479**: Added discussion on the limitations of this study.

41. **Appendix A**: Shortened appendix down by removing description of how to generate the bedrock in 3D.

42. **Appendix A**: Specified how the gradients were generated.

---

## Author Response (AR2)

Dear Dr. Johannes Fürst and Referees,

We are delighted to hear that you find our paper suitable for publication, and we thank you for the joint effort in improving its quality.

In addition to the changes proposed we went over the LST measurements in Table 2 and Table 3 more carefully and adjusted some numbers slightly. These changes were small and did not change any conclusion, but we now have full confidence in the numbers reported. Furthermore, we found a mistake in Figure 8, where we accidentally only used one nonlinear iteration. The effect of this was to alter the appearance of the instability but fortunately did not affect the time-step size at which the spurious oscillations occur.

Please find our point-by-point response below, and as before all line numbers refer to the marked-up manuscript.

Sincerely,

André Löfgren, Josefin Ahlkrona, Thomas Zwinger, Peter Råback, and Christian Helanow

**Reviwer One Comments**

No further comments.

**Comments**

**Reviewer Two Comments**

**COMMENT 1**: I noticed a typo at line 169. There is a reference to eq. (15) but I think it should be eq. (14).
**RESPONSE**: Thank you for spotting this, it is now fixed.

**COMMENT 2**: I don't like the notation $()_\Omega$ for integrals over $\Omega$, and I would have prefer to see the standard notation for dot-product used in many finite elements and analysis books. That is, I would write $(f, v)_\Omega$ for $\int_\Omega f v dx$, when the integral is indeed an L2 dot-product of $f$ and $v$. But I'm OK if the authors prefer to use their notation.
**RESPONSE**: Thank you, we now follow your suggestion and use the dot-product notation. Please see Sections 3.2 - 3.4.

**List of changes**

Below is a complete list of changes made to the manuscript.

**Requested changes**

1. **Eqs 11, 12, 15, 16 and 17**: Switched to dot-product notation.

2. **Line 138**: Removed definition of $(\cdot)_\Omega$.

3. **Line 140**: Rephrased sentence slightly.

4. **Line 160**: Wrote out integral explicitly.

5. **Line 164 and 169**: Fixed reference to Eqs. (14) and (15).

6. **Line 184**: Wrote out integral explicitly.

**Unsolicited changes**

1. **Table 2 and 3**: Updated LST values.

2. **Lines 267–271**: Slightly modified paragraph to reflect new values in Table 2.

3. **Lines 324–325**: Changed reported numbers to reflect new values in Table 3.

4. **Figure 6**: Combined panels into one figure.

5. **Figure 8**: Regenerated figure showing sloshing instability using multiple nonlinear iterations.

6. **Line 385**: Clarified where instabilities are seen in Figure 8.